# D-Separation for Causal Self-Explanation

**Wei Liu**[1]    **Jun Wang**[2*]   **Haozhao Wang**[1*]   **Ruixuan Li**[1*]
**Zhiying Deng**[1]    **Yuankai Zhang**[1]    **Yang Qiu**[1]

[1]School of Computer Science and Technology, Huazhong University of Science and Technology
[2]iWudao Tech

[1]{idc_lw, hz_wang, rxli, dengzhiyingdd, yuankai_zhang, anders}@hust.edu.cn
[2]jwang@iwudao.tech

## Abstract

Rationalization is a self-explaining framework for NLP models. Conventional work typically uses the maximum mutual information (MMI) criterion to find the rationale that is most indicative of the target label. However, this criterion can be influenced by spurious features that correlate with the causal rationale or the target label. Instead of attempting to rectify the issues of the MMI criterion, we propose a novel criterion to uncover the causal rationale, termed the Minimum Conditional Dependence (MCD) criterion, which is grounded on our finding that the non-causal features and the target label are *d-separated* by the causal rationale. By minimizing the dependence between the unselected parts of the input and the target label conditioned on the selected rationale candidate, all the causes of the label are compelled to be selected. In this study, we employ a simple and practical measure of dependence, specifically the KL-divergence, to validate our proposed MCD criterion. Empirically, we demonstrate that MCD improves the F1 score by up to 13.7% compared to previous state-of-the-art MMI-based methods. Our code is available at: `https://github.com/jugechengzi/Rationalization-MCD`.

## 1 Introduction

With the success of deep learning, there is growing concern about the interpretability of deep learning models, particularly as they are rapidly being deployed in various critical fields (Lipton, 2018). Ide-

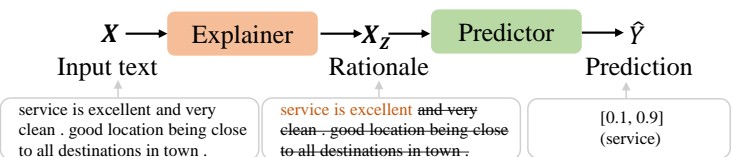

Figure 1: The standard rationalization framework RNP. $X$ is the original full text. $X_Z$ is the selected rationale candidate and $\hat{Y}$ is the predictor's output.

ally, the explanation for a prediction should be both faithful (reflecting the model's actual behavior) and plausible (aligning with human understanding) (Chan et al., 2022).

Post-hoc explanations, which are trained separately from the prediction process, may not faithfully represent an agent's decision, despite appearing plausible (Lipton, 2018). Sometimes, faithfulness should be considered a prerequisite that precedes plausibility in explanations of neural networks, especially when these networks are employed to assist in critical decision-making processes, as this factor determines the trustworthiness of the explanations. In contrast to post-hoc methods, ante-hoc (or self-explaining) techniques typically offer increased transparency (Lipton, 2018) and faithfulness (Yu et al., 2021), as the prediction is made based on the explanation itself.

---

*Corresponding authors. This paper is a collaboration between Intelligent and Distributed Computing Laboratory, Huazhong University of Science and Technology and iWudao Tech.

37th Conference on Neural Information Processing Systems (NeurIPS 2023).

A model-agnostic ante-hoc explanation framework, called Rationalizing Neural Predictions (RNP), was proposed by Lei et al. (2016) and is also known as rationalization. RNP utilizes a cooperative game between an explainer and a predictor, where the explainer identifies a human-interpretable subset of the input (referred to as rationale) and passes it to the subsequent predictor for making predictions, as shown in Figure 1. The explainer and predictor are trained cooperatively to maximize prediction accuracy. A significant advantage of RNP-based rationalization is its certification of exclusion, which guarantees that any unselected part of the input has no contribution to the prediction. This property ensures the maintenance of faithfulness, enabling us to focus solely on plausibility (Yu et al., 2021). Notably, although RNP was initially proposed in the field of NLP and and its enhancement schemes have primarily been validated using text data, its framework can also be applied to other domains, e.g., explaining image classification (Yuan et al., 2022) and graph neural networks (Luo et al., 2020).

Previous rationalization methods generally utilize the maximum mutual information (MMI) criterion to determine the rationale, defined as the subset most indicative of the target label. However, this criterion merely uncovers associations rather than causal relationships between the rationale and the label. Consequently, MMI is easily affected by spurious correlations and the plausibility of chosen rationales might be diminished, even though the rationales still faithfully report the predictor's behavior (Chang et al., 2020). In rationalization, there are two stages from which correlations may arise. The first type of correlation originates from the process of dataset generation, and we refer to it as feature correlation. A typical example of feature correlation, as pointed out in LIME (Ribeiro et al., 2016), is that wolves often appear together with snow. Consequently, whether the background features snow or not can serve as a strong indicator for classifying an image as depicting a wolf. Another instance of feature correlation is demonstrated in the first row of Table 1. Within a beer review, a favorable taste often correlates with an appealing aroma. Comments regarding the taste can serve as a strong indicator for the smell label. The predictor might inadvertently overfit to such correlations, leading to local optima. Consequently, the suboptimal predictor could mislead the explainer to select these spurious correlations. Another type of correlation stems from the rationale (mask) selection stage, and we call it mask correlation. An example of mask correlation is depicted in the second row of Table 1. Consider a situation where the explainer has implicitly learned the category of $X$, and selects a "-" for all negative inputs while excluding it from all positive inputs. In this case, the predictor only needs to determine whether the input rationale includes a "-" or not. Even though this phenomenon has not been analyzed from the perspective of spurious correlation, it has been observed and named as **degeneration** in prior research (Yu et al., 2019).

Some methods have been developed to address either feature correlation or degeneration separately. INVRAT (Chang et al., 2020) attempts to tackle feature correlation using invariant risk minimization (IRM) (Arjovsky et al., 2019). The main idea is to emphasize spurious (non-causal) variations by splitting the dataset into distinct environments. However, IRM-based methods have several limitations. For instance, they require strong prior knowledge about the relationships between non-causal and causal features (e.g., the extra labels of non-causal features) in order to divide the dataset (Lin et al., 2022b). Moreover, IRM-based methods are limited to addressing only a finite set of predetermined non-causal features, neglecting the potential existence of numerous unknown non-causal features. In fact, a recent study (Lin et al., 2022b) in the field of IRM has theoretically demonstrated that it is nearly impossible to partition a dataset into different environments to eliminate all non-causal features using IRM. Other challenges, such as the tendency to overfit, difficulty in applying to larger models (Zhou et al., 2022; Lin et al., 2022a), and the marginal shift risk of the input (Rosenfeld et al., 2021), have also been identified within the realm of IRM. Inter_RAT (Yue et al., 2023) attempts to eliminate feature correlation through *backdoor adjustment*, intervening directly with the confounders. However, it is extremely hard to measure the confounders since they are usually not observable in the dataset. As for degeneration, although not explicitly associated with spurious correlation until this study, some efforts have been tried to fix the problem. The common idea is to introduce auxiliary modules that have access the full texts to regularize the original explainer (Yu et al., 2021) or the predictor (Yu et al., 2021; Liu et al., 2022). Regularized by these auxiliary modules, the predictor can somewhat disregard the mask correlation, and degeneration is partially alleviated.

Although these methods have tried to fix the problems resulted from spurious correlations, they are still MMI-based methods and how the problems come into being is not well explored. In this study, we first identify the two stages that the spurious correlations may come from, and link the important degeneration problem to a more general mask correlation. Then, we identify that the target label $Y$ and all the non-causal features (including the rationale masks) in the input $X$ are *d-separated* by the

Table 1: The examples of feature correlation and mask correlation. Human-annotated rationales are underlined. Rationales from RNP are highlighted in red.

| RNP |
| --- |
| **Dataset:** Beer-Aroma. **Label:** Positive. **Predition:** Positive. **Problem:** feature correlation |
| **Text:** the appearance was nice . dark gold with not much of a head but nice lacing when it started to dissipate . the smell was ever so hoppy with a hint of the grapefruit flavor that 's contained within . the taste was interesting , up front tart grapefruit , not sweet in the least . more like grapefruit rind even . ⋯⋯. |
| **Dataset:** Beer-Aroma. **Label:** Negative. **Predition:** Negative. **Problem:** mask correlation |
| **Text:** 12 oz bottle poured into a pint glass - a - pours a transparent , pale golden color . the head is pale white with no cream , one finger 's height , and abysmal retention . i looked away for a few seconds and the head was gone  s - stale cereal grains dominate . hardly any other notes to speak of . very mild in strength t - sharp corn/grainy notes throughout it 's entirety . ⋯⋯. |

causal features, meaning that the non-causal features are independent of $Y$ given the causal features. This leads to a new avenue for addressing feature correlation and mask correlation simultaneously: we only need to penalize the dependence between the target label and the unselected features conditioned on the selected rationale candidate, such that all the direct causal features will be included in the selected rationale candidate. Based on this observation, we develop a new criterion for the causal rationale, namely minimum conditional dependence (MCD). Various methods can be adopted to measure the dependence, such as mutual information, the Hilbert-Schmidt Independence Criterion (HSIC) (Gretton et al., 2007), and so on. In this paper, we adopt a simple and practical measurement for independence, the KL-divergence, to verify the effectiveness of the proposed criterion. Then, we conduct experiments on two widely used benchmarks to validate the effectiveness of MCD.

In summary, our contributions are:

- To the best of our knowledge, we are the first to identify the degeneration problem as a form of spurious correlation. Leveraging probabilistic graphical models, we are the first to comprehensively elucidate feature correlation and degeneration under a unified perspective.

- We find that the target label and non-causal features are *d-separated* by the direct causal features. Based on this insight, we propose the MCD criterion, which opens a new avenue for discovering causal rationales, marking the main contribution of this study. Unlike previous methods, MCD-based methods do not require prior expert knowledge about non-causal features, thus presenting potential for broader applicability.

- We present a simple and practical architecture to develop an MCD-based method. Experiments across various datasets demonstrate that our approach achieves an improvement of up to 13.7% in F1 score compared to state-of-the-art MMI-based rationalization methods.

## 2   Related work

**Rationalization**. The basic cooperative framework of rationalization called RNP (Lei et al., 2016) is flexible and offers a unique advantage: certification of exclusion, which means any unselected input is guaranteed to have no contribution to the prediction (Yu et al., 2021). Based on this cooperative framework, many methods have been proposed to improve RNP from various aspects. Bao et al. (2018) used Gumbel-softmax to do the reparameterization for binarized selection. Bastings et al. (2019) replaced the Bernoulli sampling distributions with rectified Kumaraswamy distributions. Jain et al. (2020) disconnected the training regimes of the generator and predictor networks using a saliency threshold. Paranjape et al. (2020) imposed a discrete bottleneck objective to balance the task performance and the rationale length. Chang et al. (2019) tried to select class-wise rationales. Antognini et al. (2021); Antognini and Faltings (2021) tried to select rationales belonging to different aspects at once. Zheng et al. (2022) called for more rigorous evaluation of rationalization models. Fernandes et al. (2022) leveraged meta-learning techniques to improve the quality of the explanations. Havrylov et al. (2019) cooperatively trained the models with standard continuous and discrete optimization schemes. Hase et al. (2020) explored better metrics for the explanations. Rajagopal et al. (2021) used phrase-based concepts to conduct a self-explaining model. Other methods like data augmentation with pretrained models (Plyler et al., 2021), training with human-annotated rationales (Chan et al., 2022), have also been tried. These methods are orthogonal to our research.

**Spurious correlations**. Several methods have been proposed to address the issues arising from either feature correlation or mask correlation. The impact of feature correlation is somewhat mitigated by techniques such as invariant risk minimization (Chang et al., 2020) or backdoor adjustment (Yue et al., 2023). However, as indicated in the introduction, these methods have certain limitations. To combat mask correlation, the usual strategy involves introducing an auxiliary module, which has access to the full input, to regulate the original modules and prevent them from overfitting to trivial patterns introduced by the explainer (Yu et al., 2021, 2019; Liu et al., 2022). Other methods like using multiple explainers to select diverse rationales (Liu et al., 2023a), assigning asymmetric learning rates for the two players (Liu et al., 2023b), have also been tried. Unfortunately, these methods have limited effectiveness against feature correlation in the input data. These aforementioned methods are most relevant to our research, yet we are the first to consider both feature correlation and mask correlation from a unified perspective.

## 3    Preliminaries

We consider the text classification task, where the input is a text sequence $X=[x_1, x_2, \cdots, x_l]$ with $x_i$ being the $i$-th token and $l$ being the number of tokens. The label of $X$ is a one-hot vector $Y \in \{0,1\}^c$, where $c$ is the number of categories. $\mathcal{D}$ represents the training set. Ante-hoc rationalization consists of an explainer $f_E(\cdot)$ and a predictor $f_P(\cdot)$, with $\theta_e$ and $\theta_p$ representing the parameters of the explainer and predictor, respectively. The goal of an MMI-based explainer is to select the most indicative pieces from the input that are related to the label.

For $(X, Y) \sim \mathcal{D}$, the explainer first outputs a sequence of binary mask $M = f_E(X) = [m_1, \cdots, m_l] \in \{0,1\}^l$ (in practice, the explainer first outputs a Bernoulli distribution for each token and the mask for each token is independently sampled using gumbel-softmax). Then, it forms the rationale candidate $X_Z$ by the element-wise product of $X$ and $M$:

$$X_Z = M \odot X = [m_1 x_1, \cdots, m_l x_l]. \tag{1}$$

To simplify the notation, we denote $f_E(X)$ as $X_Z$ in the following sections, i.e., $f_E(X) = X_Z$. With the generator's selection, we get a set of $(Z, Y)$ pairs, which are generally considered to be samples taken from the distribution $P(Z, Y)$. Then, vanilla RNP attempts to identify the rationale by maximizing the mutual information $I(Y; X_Z)$:

$$X_Z^* = \arg\max_{X_Z} I(Y; X_Z) = \arg\max_{X_Z}(H(Y) - H(Y|X_Z)) = \arg\min_{X_Z} H(Y|X_Z),\ s.t.\ X_Z = f_E(X). \tag{2}$$

In practice, the entropy $H(Y|X_Z)$ is commonly approximated by the minimum cross-entropy $\min_{\theta_p} H_c(Y, \hat{Y}|X_Z)$, with $\hat{Y} = f_P(X_Z)$ representing the output of the predictor. It is essential to note that the minimum cross-entropy is equal to the entropy (please refer to Appendix B.3). Replacing $X_Z$ with $f_E(X)$, the explainer and the predictor are trained cooperatively:

$$\min_{\theta_e, \theta_p} H_c(Y, f_P(f_E(X))|f_E(X)),\ s.t.,\ (X, Y) \sim \mathcal{D}. \tag{3}$$

To make the selected rationale human-intelligible, rationalization methods usually constrain the rationales by compact and coherent regularization terms. In this paper, we use the same constraints used in INVRAT (Chang et al., 2020):

$$\Omega(M) = \lambda_1 \left| \frac{\|M\|_1}{l} - s \right| + \lambda_2 \sum_{t=2}^{l} |m_t - m_{t-1}|. \tag{4}$$

The first term encourages that the percentage of the tokens being selected as rationales is close to a pre-defined level $s$. The second term encourages the rationales to be coherent.

## 4    Method

### 4.1    Motivation: how spurious correlations come into being.

In this section, we consider $X$ as a set of variables (or a multi-dimensional variables), and the selected rationale candidate $X_Z$ is a subset (some dimensions) of it.

To begin with, in Figure 2(a), we posit a probabilistic graphical model to illustrate the corresponding data-generating process for the *BeerAdvocate* dataset. The input $X$ comprises comments on three aspects: $X_S$ for **S**mell or Aroma, $X_T$ for **T**aste, and $X_A$ for **A**ppearance, each of which can be considered as a subset variables of $X$. Additionally, $H$ signifies something that does not discuss the sentiment tendency of $X$. For instance, $H$ could include the color of a bottle. The annotators assign the smell label $Y_S$ by viewing the comments on aroma ($X_S \rightarrow Y_S$). Therefore, only $X_S$ serves as the direct cause for $Y_S$. However, $X_S$ is correlated with $X_T$ due to a set of unobserved variables $U$ (called *confounders*). For example, $U$ may include a variable indicating whether the beer originates from a reputable brand, and a pleasant taste may imply that the beer comes from a good brand ($U \rightarrow X_T$). Moreover, a beer from a reputable brand is likely to have a pleasing smell ($U \rightarrow X_S$). Consequently, $X_T$ is associated with $Y_S$ via a *backdoor* path, as depicted by the red dotted line in Figure 2(a). In this situation, $X_T$ is somewhat indicative of $Y_S$, but it signifies a statistical correlation rather than causality.

To have a more intuitive understanding of this correlation, we assume a toy example where $U$, $X_S$, $X_T$, and $Y_S$ are all Bernoulli variables, with their respective probability distributions as:

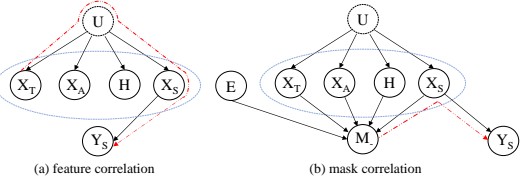

(a) feature correlation      (b) mask correlation

$p(U = 1) = p(U = 0) = 0.5,$

$p(X_T = 1|U = 1) = p(X_T = 0|U = 0) = 0.9,$

$p(X_S = 1|U = 1) = p(X_S = 0|U = 0) = 0.9,$

$p(Y_S = 1|X_S = 1) = p(Y_S = 0|X_S = 0) = 0.9,$ 

(5)

Figure 2: A probabilistic graph for (a) the data generating process of *BeerAdvocate* and (b) the rationale (mask $M$) selection process.

With some simple derivations, we can easily obtain (detailed derivation is in Appendix B.1):

$$p(X_S = 1) = p(X_T = 1) = p(Y_S = 1) = 0.5. \qquad (6)$$

Then, we can further get (see Appendix B.2 for the detailed derivation of Equation 8 and 9):

$$p(U = 1|X_T = 1) = \frac{p(U = 1, X_T = 1)}{p(X_T = 1)} = \frac{p(X_T = 1|U = 1)p(U = 1)}{p(X_T = 1)} = 0.9. \qquad (7)$$

$$p(X_S = 1|X_T = 1) = \sum_{U \in \{0,1\}} p(X_S = 1|U)p(U|X_T = 1) = 0.9 * 0.9 + 0.1 * 0.1 = 0.82. \qquad (8)$$

$$p(Y_S = 1|X_T = 1) = \sum_{X_S \in \{0,1\}} p(Y_S = 1|X_S)p(X_S|X_T = 1) = 0.82 * 0.9 + 0.18 * 0.1 = 0.756. \qquad (9)$$

Equation 8 demonstrates that $X_T$ (Taste) is highly correlated with $X_S$ (Smell), and Equation 9 indicates that $X_T$ (Taste) is also strongly indicative of $Y_S$ (Smell label). This situation can result in numerous local optima during the rationalization training process. Note that becoming trapped in a local optimum poses a significant challenge in rationalization (Yu et al., 2021; Liu et al., 2022). It is worth noting that the correlation between taste and smell here is merely one of the examples, and sometimes $H$ can also correlate with $Y$ in a similar fashion. For instance, in LIME (Ribeiro et al., 2016), a predictor is trained to determine whether an image contains a wolf or not based on the presence of snow in the background.

Furthermore, the degeneration problem can also be interpreted with a kind of spurious correlation (called mask correlation), as illustrated in Figure 2(b) with an example. Returning to the second example in Table 1, the variable $M_-$, denoting whether " – " is selected as part of the rationale candidate, is caused by the input $X$ (comprising subsets $X_A$, $X_T$, $X_S$ and $H$) and the explainer $E$. $M_-$ is also correlated with $Y_S$ through a backdoor path, as indicated by the red line in Figure 2(b).

## 4.2 The conditional independent property

We first introduce an important concept in probabilistic graphical models, namely **d-separation**. Subsequently, we demonstrate how d-separation contributes to the identification of causal rationales.

**D-Separation** (Bishop, 2006): $A$, $B$, and $C$ denote arbitrary, non-intersecting sets of nodes (and their union might not cover all nodes of the graph) in a given probabilistic graph. Our objective is to determine whether a specific conditional independence statement $A \perp\!\!\!\perp B|C$ is implied by this graph. To do so, we examine all possible paths from any node in $A$ to any node in $B$. A path is said to be blocked if it includes a node $o$ such that either (see Appendix B.4 for why such a path is blocked)

- (a) The arrows on the path meet at node $o$, forming either a chain (i.e., $\to o \to$) or a fork (i.e., $\leftarrow o \to$), with the node $o$ being part of set C, or
- (b) The arrows on the path meet at node $o$ to form a collider (i.e., $\to o \leftarrow$), and neither the node $o$ itself nor any of its descendants are included in set C.

If all paths are blocked, then $A$ is considered to be **d-separated** from $B$ by $C$, meaning that $A \perp\!\!\!\perp B|C$.

Returning to our rationalization problem, the backdoor path (dotted red line) in Figure 2(a) comprises a fork ($X_T \leftarrow U \to X_S$) and a chain ($U \to X_S \to Y_S$). If either $X_S$ or $U$ is included in the conditioning set, the path between $X_S$ and $Y_S$ becomes blocked, leading to their conditional independence, and consequently, the eradication of corresponding feature correlation. Similarly, the backdoor path (dotted red line) in Figure 2 (b) forms a fork ($M_- \leftarrow X_S \to Y_S$). By including $X_S$ in the conditioning set, the path between $M_-$ and $Y_S$ is blocked, resulting in their conditional independence and consequently, the elimination of corresponding mask correlation.

We consider the general case, where the input $X$ is a set of variables (or features). $X_R$ is a subset of $X$ that exclusively contains all the direct causes of the target label $Y$, i.e., the desiderata of the rationale. We select a subset of $X$ to serve as the rationale candidate (denoted as $X_Z$), while the remaining unselected part is referred to as $X_{-Z}$. This leads us to the following properties:

**Lemma 1** *If $X_{-Z}$ and $Y$ are d-separated by $X_Z$, we then have that all of the direct causal features in $X$ must be included in $X_Z$:*

$$X_{-Z} \text{ and } Y \text{ are d-separated by } X_Z \Longrightarrow X_R \subset X_Z. \tag{10}$$

The proof is in Appendix B.5. And it's very easy to intuitively understand it: a direct cause has a one-hop path to the label. To block this path, this cause must be included in $X_Z$.

**Assumption 1** *The label $Y$ has no causal effect on any variables in $X$.*

Assumption 1 is naturally valid in most real world applications due to the temporal sequence between $X$ and $Y$. We also provide some failure cases of this assumption in Appendix B.6. Assumption 1 specifies that there is no arrow pointing from $Y$ to any nodes in $X$.

**Lemma 2** *If Assumption 1 holds, we have:*

$$X_{-Z} \text{ and } Y \text{ are d-separated by } X_Z \Longleftarrow X_R \subset X_Z. \tag{11}$$

The proof is in Appendix B.7. Combining Lemma 1 and Lemma 2, we then have:

**Theorem 1** *If Assumption 1 holds, then all the direct causal features to $Y$ within $X$ will be included in $X_Z$ if and only if $X_{-Z}$ and $Y$ are d-separated by $X_Z$:*

$$X_{-Z} \text{ and } Y \text{ are d-separated by } X_Z \Longleftrightarrow X_R \subset X_Z. \tag{12}$$

**Remark**. In light of Theorem 1, we understand that if we aim to achieve $Y \perp\!\!\!\perp X_{-Z}|X_Z$, we will consequently incorporate all direct causes of $Y$ into $X_Z$. It should be noted that the compactness of $X_Z$ is facilitated through the sparsity constraint expressed in Equation 4.

### 4.3 The proposed method

**Minimum conditional dependence criterion**. Although previous research tried to design various auxiliary modules or regularizers to fix the problems of maximum mutual information criterion (Chang et al., 2020; Yue et al., 2023; Liu et al., 2022), we do not follow them to move on this line. Based on Theorem 1, we propose a distinct criterion for identifying the causal rationale, which involves minimizing the dependence between $Y$ and the unselected input $X_{-Z}$, conditioned on $X_Z$:

$$X_Z^* = \arg\min_{X_Z} \mathcal{C}(Y, X_{-Z}|X_Z), \tag{13}$$

where $\mathcal{C}$ is a criterion for dependence. For instance, $\mathcal{C}$ could take the form of partial correlation (applicable only to linear associations), mutual information, divergence, or the Hilbert-Schmidt Independence Criterion (HSIC) (Gretton et al., 2007), among others.

It then leads to the question of how we can apply this criterion in practice. In this study, we only present a straightforward and practical method to validate our assertion with respect to Theorem 1, leaving the exploration of other measurements for future work. We first rewrite $Y \perp\!\!\!\perp X_{-Z}|X_Z$ as

$$P(Y|X_Z) = P(Y|X_Z, X_{-Z}) = P(Y|X). \tag{14}$$

Figure 3: The architecture of our proposed MCD. The approximators for the two distributions are shared to reduce the model complexity (this trick is not necessary).

Obviously, $P(Y|X_Z) = P(Y|X)$ if and only if the divergence between the two distributions is zero:

$$Y \perp\!\!\!\perp X_{-Z}|X_Z \iff P(Y|X_Z) = P(Y|X) \iff D_{KL}(P(Y|X)\|P(Y|X_Z)) = 0. \tag{15}$$

**Estimating divergence through approximation**. The real distributions of $P(Y|X_Z)$ and $P(Y|X)$ are not directly accessible. So we need further efforts to approximate them. We try to approximate them by making use of the predictor. We first approximate $P(Y|X_Z)$ with $P(\hat{Y}_Z|X_Z)$ by minimizing the cross-entropy $H_c(Y, \hat{Y}_Z|X_Z)$, and we also approximate $P(Y|X)$ with $P(\hat{Y}_X|X)$ by minimizing $H_c(Y, \hat{Y}_X|X)$, where $\hat{Y}_Z, \hat{Y}_X$ are the predictor's outputs with the inputs being $Z$ and $X$, respectively.

Thus, the training process for our MCD is depicted in Figure 3: the explainer first generates a rationale candidate $X_Z$ from the input $X$. Subsequently, $X_Z$ and $X$ are fed into the predictor to obtain two distributions, $P(\hat{Y}_Z|X_Z)$ and $P(\hat{Y}_X|X)$. By replacing $X_Z$ with $f_E(X)$ and $\hat{Y}$ with $f_P(\cdot)$, the overall objective of our model becomes (The pytorch implementation is in Appendix A.2):

$$\min_{\theta_p} E_{(X,Y)\sim\mathcal{D}}[H_c(Y, \hat{Y}_Z|X_Z) + H_c(Y, \hat{Y}_X|X)]$$
$$+ \min_{\theta_e} E_{(X,Y)\sim\mathcal{D}}[D_{KL}(P(\hat{Y}_X|X)\|P(\hat{Y}_Z|X_Z)) + \Omega(M)], \tag{16}$$
$$s.t., \ X_Z = f_E(X), \ P(\hat{Y}_X|X) = f_P(X), \ P(\hat{Y}_Z|X_Z) = f_P(X_Z).$$

Notably, although the first term $H_c(Y, \hat{Y}_Z|X_Z)$ is similar to the one used in Equation 3, it is detached from the explainer's parameters $\theta_e$. It is now only used to help the predictor approximate the real distribution $P(Y|X_Z)$ rather than to guide the explainer to find a good rationale.

## 5 Experiments

### 5.1 Datasets and metrics

**Datasets** 1) **BeerAdvocate** (McAuley et al., 2012) is a multi-aspect sentiment prediction dataset widely adopted in rationalization studies. Given the high correlation among the rating scores of different aspects within the same review, rationale selection encounters severe feature correlation challenges. Following INVRAT (Chang et al., 2020) and Inter_RAT (Yue et al., 2023), we utilize the original dataset (which we refer to as *correlated BeerAdvocate*) to verify MCD's effectiveness in handling both feature correlation and mask correlation simultaneously. 2) **HotelReviews** (Wang et al., 2010) is another multi-aspect sentiment classification dataset containing less feature correlation, which is used by the latest SOTA method FR (Liu et al., 2022) to evaluate the effectiveness of addressing degeneration. We utilize the *Service* aspect to further demonstrate the competitive edge of our MCD. Among these datasets, each aspect itself can be seen as a dataset and is trained independently.

**Metrics**. Considering that the annotators assign the label of the target aspect by observing the causal features, the overlap between the tokens selected by the model and those annotated by humans provides a robust metric for rationale causality. The terms $P, R, F1$ denote precision, recall, and $F1$ score respectively. These metrics are the most frequently used in rationalization. The term $S$ represents the average sparsity of the selected rationales, that is, the percentage of selected tokens in relation to the full text. $Acc$ stands for the predictive accuracy.

### 5.2 Baselines and implementation details

We compare with various recent MMI-based methods that are highly relevant to our study. These include methods like INVRAT (Chang et al., 2020) and Inter_RAT (Yue et al., 2023), which are

Table 2: Results on *correlated BeerAdvocate*. Each aspect is trained independently. " ∗ ": results obtained from Inter_RAT (Yue et al., 2023). The second best F1 scores are underlined.

| Methods | Appearance | | | | | Aroma | | | | | Palate | | | | |
|---|---|---|---|---|---|---|---|---|---|---|---|---|---|---|---|
| | S | Acc | P | R | F1 | S | Acc | P | R | F1 | S | Acc | P | R | F1 |
| RNP* | 10.0 | - | 32.4 | 18.6 | 23.6 | 10.0 | - | 44.8 | 32.4 | 37.6 | 10.0 | - | 24.6 | 23.5 | 24.0 |
| INVRAT* | 10.0 | - | 42.6 | 31.5 | 36.2 | 10.0 | - | 41.2 | 39.1 | 40.1 | 10.0 | - | 34.9 | 45.6 | 39.5 |
| Inter-RAT* | 11.7 | - | 66.0 | 46.5 | 54.6 | 11.7 | - | 55.4 | 47.5 | 51.1 | 12.6 | - | 34.6 | 48.2 | 40.2 |
| FR | 11.1 | 75.8 | 70.4 | 42.0 | 52.6 | 9.7 | 87.7 | 68.1 | 42.2 | 52.1 | 11.7 | 87.9 | 43.7 | 40.9 | 42.3 |
| MCD(ours) | 9.5 | 81.5 | **94.2** | **48.4** | **63.9** | 9.9 | 87.5 | **84.6** | **53.9** | **65.8** | 9.4 | 87.3 | **60.9** | **47.1** | **53.1** |
| RNP* | 20.0 | - | 39.4 | 44.9 | 42.0 | 20.0 | - | 37.5 | 51.9 | 43.5 | 20.0 | - | 21.6 | 38.9 | 27.8 |
| INVRAT* | 20.0 | - | 58.9 | 67.2 | 62.8 | 20.0 | - | 29.3 | 52.1 | 37.5 | 20.0 | - | 24.0 | 55.2 | 33.5 |
| Inter-RAT* | 21.7 | - | 62.0 | 76.7 | 68.6 | 20.4 | - | 44.2 | 65.4 | 52.8 | 20.8 | - | 26.3 | 59.1 | 36.4 |
| FR | 20.9 | 84.6 | 74.9 | 84.9 | 79.6 | 19.5 | 89.3 | 58.7 | 73.3 | 65.2 | 20.2 | 88.2 | 36.6 | 59.4 | 45.3 |
| MCD(ours) | 20.0 | 85.5 | **79.3** | **85.5** | **82.3** | 19.3 | 88.4 | **65.8** | **81.4** | **72.8** | 19.6 | 87.7 | **41.3** | **65.0** | **50.5** |
| RNP* | 30.0 | - | 24.2 | 41.2 | 30.5 | 30.0 | - | 27.1 | 55.7 | 36.4 | 30.0 | - | 15.4 | 42.2 | 22.6 |
| INVRAT* | 30.0 | - | 41.5 | 74.8 | 53.4 | 30.0 | - | 22.8 | 65.1 | 33.8 | 30.0 | - | 20.9 | 71.6 | 32.3 |
| Inter-RAT* | 30.5 | - | 48.1 | 82.7 | 60.8 | 29.4 | - | 37.9 | 72.0 | 49.6 | 30.4 | - | 21.8 | 66.1 | 32.8 |
| FR | 29.6 | 86.4 | 50.6 | 81.4 | 62.3 | 30.8 | 88.1 | 37.4 | 75.0 | 49.9 | 30.1 | 87.0 | 24.5 | 58.8 | 34.6 |
| MCD(ours) | 29.7 | 86.7 | **59.6** | **95.6** | **73.4** | 29.6 | 90.2 | **46.1** | **87.5** | **60.4** | 29.4 | 87.0 | **30.5** | **72.4** | **42.9** |

focused on addressing feature correlation, as well as methods such as FR (Liu et al., 2022) that aim to mitigate mask correlation (i.e., degeneration). Among these, FR represents the latest SOTA approach in addressing mask correlation, while Inter_RAT stands as the SOTA in handling feature correlation.

Both the explainer and the predictor are composed of an encoder (which can be an RNN or Transformer) and a linear layer. Some of the baseline methods have not provided runnable source codes. To ensure a fair comparison, we keep the major settings consistent with those of the baselines, which are commonly utilized in the field of rationalization (Chang et al., 2020; Yu et al., 2021; Liu et al., 2022; Yue et al., 2023). Specifically, we employ the 100-dimensional GloVe (Pennington et al., 2014) for word embedding and 200-dimensional GRUs (Cho et al., 2014) to obtain text representation. The re-parameterization trick for binarized selection is Gumbel-softmax (Jang et al., 2017). The hyperparameters of the reimplemented baselines are initialized with the values reported in their source codes, and are then manually tuned multiple times to determine the optimal settings. We do not use BERT (Devlin et al., 2019) in the main experiments because some recent research (Chen et al., 2022; Liu et al., 2022; Zhang et al., 2022) has found it to be a challenging task to fine-tune large pretrained models within the rationalization framework (see Appendix A.4 for more discussion). However, as a supplement, we also conduct experiments with two pretrained models, ELECTRA (Clark et al., 2020) and BERT. The optimizer is Adam (Kingma and Ba, 2015). All models are trained on a RTX3090 GPU. More details are in Appendix A.1.

## 5.3 Results

**Comparison with SOTA Methods**. Table 2 shows the results on *correlated BeerAdvocate* with the rationale sparsity being about $10\%$, $20\%$, and $30\%$. We set the sparsity to be similar to previous methods by adjusting the sparsity regularization term (i.e., $s$) in Equation 4. Compared to MMI-based methods, we gain significant improvements across all three aspects and three different sparsity. In particular, we improve

Table 3: Results on *HotelReview*. "*": results obtained from FR (Liu et al., 2022).

| Methods | S | Acc | P | R | F1 |
|---|---|---|---|---|---|
| RNP* | 11.0 | 97.5 | 34.2 | 32.9 | 33.5 |
| DMR* | 11.6 | - | 43.0 | 43.6 | 43.3 |
| A2R* | 11.4 | 96.5 | 37.3 | 37.2 | 37.2 |
| FR* | 11.5 | 94.5 | 44.8 | 44.7 | 44.8 |
| MCD(ours) | 11.8 | 97.0 | **47.0** | **48.6** | **47.8** |

the F1 score by more then $10\%$ as compared to the previous SOTA in three settings: in the *Aroma* aspect with $S \approx 10$, the *Palate* aspect with $S \approx 10$, and the *Appearance* aspect with $S \approx 30$. We show an visualized example of the selected rationales in Figure 4. Since our MCD criterion (Equation 13) is not limited to a specific measurement of dependence, we also conduct experiments by replacing KL-divergence with JS-divergence, and the results are in Appendix A.5. Table 3 shows the results on another dataset also used in FR, where DMR (Huang et al., 2021) and A2R (Yu et al., 2021) are two recent MMI-based methods. For this dataset, we follow FR to set the sparsity similar to that of the human-annonated rationales. On this dataset, we still beat all the MMI-based methods. We also show the time efficiency in Appendix A.6.

**Inducing mask correlation with skewed explainer**. In order to evaluate scenarios where feature correlation is not severe and our primary concern is mask correlation, we follow FR's approach to conduct experiments in a synthetic setting where the explainer is specifically initialized to induce

Table 4: Results of skewed explainer that induces degeneration (i.e., mask correlation) in the *Palate* aspect of *BeerAdvocate*. "$*$": results obtained from the paper of FR.

| Setting | RNP* | | | | | | FR* | | | | | | MCD(ours) | | | | | |
|---|---|---|---|---|---|---|---|---|---|---|---|---|---|---|---|---|---|---|
| | Pre_acc | S | Acc | P | R | F1 | Pre_acc | S | Acc | P | R | F1 | Pre_acc | S | Acc | P | R | F1 |
| skew65.0 | 66.6 | 14.0 | 83.9 | 40.3 | 45.4 | 42.7 | 66.3 | 14.2 | 81.5 | 59.5 | **67.9** | **63.4** | 66.3 | 12.9 | 84.6 | **61.6** | 63.7 | 62.6 |
| skew70.0 | 71.3 | 14.7 | 84.1 | 10.0 | 11.7 | 10.8 | 70.8 | 14.1 | 88.3 | 54.7 | 62.1 | 58.1 | 70.2 | 13.5 | 81.1 | **59.0** | **64.0** | **61.4** |
| skew75.0 | 75.5 | 14.7 | 87.6 | 8.1 | 9.6 | 8.8 | 75.6 | 13.1 | 84.8 | 49.7 | 52.2 | 51.0 | 75.3 | 13.4 | 84.2 | **61.3** | **65.1** | **63.1** |

Table 5: Results of methods using pretrained ELECTRA as the encoder.

| Methods | Appearance | | | | | Aroma | | | | | Plate | | | | |
|---|---|---|---|---|---|---|---|---|---|---|---|---|---|---|---|
| | S | Acc | P | R | F1 | S | Acc | P | R | F1 | S | Acc | P | R | F1 |
| FR-ELECTRA | 16.3 | 86.5 | 19.1 | 17.0 | 18.0 | 14.8 | 85.9 | 58.6 | 54.8 | 56.7 | 11.2 | 78.0 | 12.0 | 10.7 | 11.3 |
| MCD-ELECTRA | 18.5 | 90.0 | **84.8** | **85.6** | **85.2** | 14.5 | 86.6 | **86.2** | **78.7** | **82.3** | 12.1 | 85.0 | **63.0** | **60.3** | **61.6** |

mask correlation, also referred to as degeneration. The details of the initialization can be found in Appendix A.3. Following FR, we utilize the *Palate* aspect of *decorrelated BeerAdvocate* dataset (a subset of the original *BeerAdvocate* that has been filtered by Lei et al. (2016)). This subset contains less feature correlation compared to the original dataset. The results are presented in Table 4, where skew$k$ and $Pre\_acc$ indicates the degree of mask correlation. In this situation, the vanilla RNP fails to identify the causal rationales, and FR is also significantly impacted when the degree of mask correlation is high. Our MCD is much less affected, demonstrating its robustness in such scenarios.

**Experiments with pretrained language models**. In the field of rationalization, researchers generally focus on frameworks of the models and the methodology rather than engineering SOTA. The methods most related to our work do not use BERT or other pre-trained encoders (Chang et al., 2020; Yu et al., 2021; Liu et al., 2022; Yue et al., 2023). Experiments in some recent work (Chen et al., 2022; Liu et al., 2022) suggest that there are some unforeseen obstacles making it hard to finetune large pretrained models within the rationalization framework. For example,

Table 6: The F1 scores of models trained with different encoders. "*": results obtained from (Chen et al., 2022). "**": results obtained from FR. The dataset is decorrelated *Beer-Appearance*.

| Method | GRU | ELECTRA | BERT |
|---|---|---|---|
| VIB* | - | - | 20.5 |
| SPECTRA* | - | - | 28.6 |
| RNP** | 72.3 | 13.7 | 14.7 |
| FR** | 82.8 | 14.6 | 29.8 |
| MCD(ours) | 80.1 | 85.2 | 87.1 |

Table 6 shows that two improved rationalization methods (VIB (Paranjape et al., 2020) and SPECTRA (Guerreiro and Martins, 2021)) and the latest published FR all fail to find the informative rationales when replacing GRUs with pretrained BERT. To eliminate potential factors that could lead to an unfair comparison, we adopt the most widely used GRUs as the encoders in our main experiments, which can help us focus more on substantiating our claims themselves, rather than unknown tricks. But to show the competitiveness of our MCD, we also provide some experiments with pretrained language models as the supplement. Due to limited GPU resources, we adopt the relatively small ELECTRA-small in all three aspects of *BeerAdvocate* and the relatively large BERT-base in the *Appearance* aspect. We compare our MCD with the latest SOTA FR (Liu et al., 2022). We follow FR to set the sparsity similar to human-annotated rationales. More details are in Appendix A.4.

The results with BERT are shown in Table 6 and results with ELECTRA are shown in Table 5. We see that our method can greatly benefit from pretrained models. In fact, recent research has found that finetuning large pretrained models can be easily affected by overfitting (Zhang et al., 2021), and spurious correlations can exacerbate this overfitting, particularly in larger models (Zhou et al., 2022; Lin et al., 2022a), which somewhat explains the great progress achieved by our MCD.

# 6 Conclusion, future work, and limitations

In this study, we first illustrate the two primary issues of feature correlation and degeneration in MMI-based rationalization under a unified causal perspective. Subsequently, we uncover the conditional independence relationship between the target label and non-causal and causal features. Based on this observation, we propose a criterion of minimizing conditional dependence to concurrently address the two aforementioned problems.

Given the versatility of the self-explaining rationalization framework, our proposed methods show significant potential for application across diverse fields such as computer vision and graph learning. Additionally, with the recent remarkable success of large language models (LLMs), exploring how our MCD can aid in training trustworthy LLMs is another avenue worth pursuing.

**Label** (aroma aspect)**:** Positive.
**Prediction:** Positive.
**Input:** got this one on tap at kelly 's olympian in portland . lighting made the colour difficult to ascertain , but i would be surprised if it were n't a very dark brown . came with a good head which stuck around , adding to the feel of the pint . i 'm useless on the sniff test in these smoky bars , so all i could distinguish was a malty cherry whiff . felt very smooth and went down very easily . was sweeter and more syrupy than i would expect an english brown ale to be . the smell did prove indicative of the initial taste , although where i expected it to give way to a somewhat hoppy finish , a thankfully subtle coffee flavour kicked in instead . i detest coffee , but this was subtle enough to be tolerable .

(a) RNP

**Label** (aroma aspect)**:** Positive.
**Prediction:** Positive.
**Input:** got this one on tap at kelly 's olympian in portland . lighting made the colour difficult to ascertain , but i would be surprised if it were n't a very dark brown . came with a good head which stuck around , adding to the feel of the pint . i 'm useless on the sniff test in these smoky bars , so all i could distinguish was a malty cherry whiff . felt very smooth and went down very easily . was sweeter and more syrupy than i would expect an english brown ale to be . the smell did prove indicative of the initial taste , although where i expected it to give way to a somewhat hoppy finish , a thankfully subtle coffee flavour kicked in instead . i detest coffee , but this was subtle enough to be tolerable .

(b) Inter_RAT

**Label** (aroma aspect)**:** Positive.
**Prediction:** Positive.
**Input:** got this one on tap at kelly 's olympian in portland . lighting made the colour difficult to ascertain , but i would be surprised if it were n't a very dark brown . came with a good head which stuck around , adding to the feel of the pint . i 'm useless on the sniff test in these smoky bars , so all i could distinguish was a malty cherry whiff . felt very smooth and went down very easily . was sweeter and more syrupy than i would expect an english brown ale to be . the smell did prove indicative of the initial taste , although where i expected it to give way to a somewhat hoppy finish , a thankfully subtle coffee flavour kicked in instead . i detest coffee , but this was subtle enough to be tolerable .

(c) FR

**Label** (aroma aspect)**:** Positive.
**Prediction:** Positive.
**Input:** got this one on tap at kelly 's olympian in portland . lighting made the colour difficult to ascertain , but i would be surprised if it were n't a very dark brown . came with a good head which stuck around , adding to the feel of the pint . i 'm useless on the sniff test in these smoky bars , so all i could distinguish was a malty cherry whiff . felt very smooth and went down very easily . was sweeter and more syrupy than i would expect an english brown ale to be . the smell did prove indicative of the initial taste , although where i expected it to give way to a somewhat hoppy finish , a thankfully subtle coffee flavour kicked in instead . i detest coffee , but this was subtle enough to be tolerable .

(d) MCD

Figure 4: An example of selected rationales in the *Aroma* aspect of BeerAdvocate. The sparsity is set to be about $10\%$. The underlined texts are human-annotated rationales. (a): RNP selects palate only. (b): Inter_RAT selects aroma but also palate ("felt very smooth"). (c): FR is similar to Inter_RAT. (d): MCD selects aroma only.

A potential limitation is that, similar to IRM-based methods, our primary focus is on identifying rationales with causal effects, rather than quantitatively computing the precise values of these causal effects. Although quantifying causality often relies on strong assumptions, this quantification may be a desideratum for certain applications. We aim to explore this direction in future work to accommodate a wider range of applications. Another limitation is that we focus on the text classification task. Different tasks may have very different causal structures. Thus, how to extend this method to other tasks is also a challenge that needs to be explored. The third limitation is that the obstacles in utilizing powerful pretrained language models under the rationalization framework remain mysterious. Although we have made some progress in this direction, we have to say that the empirical results with pretrained models are very sensitive to hyperparameter tuning. A recent paper has also shown that very small changes in hyperparameters can lead to significant differences in results (see Remark 6.1 and Appendix G.2 in (Zhang et al., 2023)). To avoid being distracted by irrelevant factors, until this issue is resolved, we call for research papers to use small models to better verify their claims.

# 7 Acknowledgements

This work is supported by National Natural Science Foundation of China under grants 62376103, 62302184, 62206102, and Science and Technology Support Program of Hubei Province under grant 2022BAA046. We are also grateful for the valuable suggestions provided by the anonymous reviewers, which greatly helped to improve the quality of this paper.

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
