Table 7: Statistics of datasets used in this paper. *: the decorrelated BeerAdvocate.

| Datasets | | Train | | Dev | | Annotation | | |
|---|---|---|---|---|---|---|---|---|
| | | Pos | Neg | Pos | Neg | Pos | Neg | Sparsity |
| Beer | Appearance | 202385 | 12897 | 28488 | 1318 | 923 | 13 | 18.5 |
| | Aroma | 172299 | 30564 | 24494 | 3396 | 848 | 29 | 15.6 |
| | Palate | 176038 | 27639 | 24837 | 3203 | 785 | 20 | 12.4 |
| Beer* | Appearance | 16891 | 16891 | 6628 | 2103 | 923 | 13 | 18.5 |
| | Aroma | 15169 | 15169 | 6579 | 2218 | 848 | 29 | 15.6 |
| | Palate | 13652 | 13652 | 6740 | 2000 | 785 | 20 | 12.4 |
| Hotel | Location | 7236 | 7236 | 906 | 906 | 104 | 96 | 8.5 |
| | Service | 50742 | 50742 | 6344 | 6344 | 101 | 99 | 11.5 |
| | Cleanliness | 75049 | 75049 | 9382 | 9382 | 99 | 101 | 8.9 |

## A  More Results

### A.1  More implementation details

To the best of our knowledge, both datasets are sufficiently anonymized to make identification of individuals impossible without significant effort. Both datasets are in English. For *correlated BeerAdvocate*, we preprocess the data in the same way as Inter_RAT (Yue et al., 2023). For *Hotel Reviews*, we preprocess them in the same way as FR (Liu et al., 2022). The maximum text length is set to 256. More statistics of the datasets are in Table 7. The dataset of *BeerAdvocate* is unbalanced. For the training data, we sample from the positive data to get same number of positive and negative texts.

In practice, the approximators for the two distributions are shared to reduce model complexity (Figure 3). But this trick is not necessary, if two separate nets are used to approximate the two distributions, the performance can sometimes be even better.

Some previous methods needs very careful hyper-parameter tuning. To make fair comparisons, most results of the baselines are copied from previous papers.

The early stopping technique is conducted according to the predictive accuracy of the development set.

For *BeerAdvocate*, we use a learning rate of 0.0001 and a batchsize of 128 for our MCD. For *HotelReview*, we use a learning rate of 0.0001 and a batchsize of 256.

We report the average results of our MDC by running it with five different random seeds.

### A.2  Pytorch implementation of Equation 16

For a batch of $(X, Y)$, we first send $X$ to the explainer to get $X_Z$:

$$X_Z = f_e(X). \tag{17}$$

Then we get a copy of $X_Z$ with the pytorch function "torch.detach()":

$$X'_Z = \text{torch.detach}(X_Z). \tag{18}$$

Then, we get $\hat{Y}_X$ and $\hat{Y}'_Z$:

$$\begin{aligned} \hat{Y}_X &= f_p(X), \\ \hat{Y}'_Z &= f_p(X'_Z). \end{aligned} \tag{19}$$

Then we update the predictor with

$$\min_{\theta_p}[\text{torch.nn.functional.cross\_entropy}(\hat{Y}'_Z, Y) + \text{torch.nn.functional.cross\_entropy}(\hat{Y}_X, Y)], \tag{20}$$

which is the first part of Equation 16. At the same time, we update the explainer with Equation 4.

Now, we deal with the second part of Equation 16. We first freeze the predictor's parameters and get $X_Z$ again:

$$X_Z = f_e(X). \tag{21}$$

We now do not copy $X_Z$. Instead, we directly get $\hat{Y}_X$ and $\hat{Y}_Z$:

$$\hat{Y}_X = f_p(X),$$
$$\hat{Y}_Z = f_p(X_Z). \tag{22}$$

Then we update the explainer with

$$\min_{\theta_e} \text{F.kl\_div}(\text{F.softmax}(\hat{Y}_Z).log(), \text{F.softmax}(\hat{Y}_X)), \tag{23}$$

where "F" denotes "nn.functional". In practice, we have added Equation 4 to 23.

Now, an update round for Equation 16 is completed, and we repeat the above steps again.

## A.3 Details of the skewed explainer

We pretrain the explainer separately using the text classification label as the mask label of the first token. In other words, for texts of class 1, we force the explainer to select the first token, and for texts of class 0, we force the explainer not to select the first token. So, the explainer learns the category implicitly by whether the first token is chosen and the predictor only needs to learn this position information to make a correct prediction.

$k$ in "skew$k$" denotes the threshold of the skew: we pretrain the explainer as a special classifier of the first token for a few epochs until its prediction accuracy is higher than $k$. Since the accuracy increases rapidly in the first a few epochs, obtaining a model that precisely achieves the pre-defined accuracy is almost impossible. So, we use "$Pre\_acc$" to denote the actual prediction accuracy of the explainer-classifier when the pre-training process stops. Higher "$Pre\_acc$" means easier to degenerate.

## A.4 Discussion on BERT encoder

In the field of rationalization, researchers generally focus on frameworks of the models and the methodology. Methods most related to our work do not use Bert or other pre-trained encoders (Chang et al., 2020; Huang et al., 2021; Yu et al., 2019, 2021; Yue et al., 2023). We use GRUs and GloVe to ensure the same experimental setup as our baselines for a fair comparison.

Table 8: Results with BERT. VIB: Paranjape et al. (2020), SPECTRA: Guerreiro and Martins (2021). The results are from Table 4 of (Chen et al., 2022). The metric is F1 score.

| Methods | Beer-Appearance | Hotel-Cleanliness |
|---------|-----------------|-------------------|
| VIB | 20.5 | 23.5 |
| SPECTRA | 28.6 | 19.5 |

More importantly, how to finetune large models on the rationalization framework is still a significant challenge. Some recent studies (Chen et al., 2022) show that the methods with BERT encoders perform much worse than those with simple GRUs on BeerAdvocate and HotelReviews, which is shown in Table 8. VIB and SPECTRA are two RNP-based models. When using BERT, these two methods perform much worse than the vanilla RNP with GRUs. Table 9 shows the results of a recent workshop paper CR (Zhang et al., 2022), which are also much worse than those with GRUs.

We also conduct experiments with pretrained language models and compare with previous methods. As previous methods are not designed to address feature correlations in the original dataset, they typically utilize the *decorrelated BeerAdvocate*) dataset where feature correlation

Table 9: The F1 scores of CR (Zhang et al., 2022) with pretrained BERT on *BeerAdvocate*. The results are from Table 1 of (Zhang et al., 2022).

| Method | Appearance | Aroma | Palate |
|--------|-----------|-------|--------|
| CR | 27.4 | 39.0 | 22.6 |

is manually filtered by Lei et al. (2016), focusing mainly on mask correlation. Following previous methods (Chen et al., 2022; Liu et al., 2022; Zhang et al., 2022), we use the *decorrelated BeerAdvocate* dataset. And we set the rationale sparsity to be similar to that of human-annotated rationales. The results are in Table 6 and Table 5.

## A.5 Experiments with JS-divergence

Table 10: Results on *correlated BeerAdvocate*. Each aspect is trained independently. "$*$": results obtained from Inter_RAT (Yue et al., 2023). The second best F1 scores are underlined.

| Methods | Appearance | | | | | Aroma | | | | | Palate | | | | |
|---|---|---|---|---|---|---|---|---|---|---|---|---|---|---|---|
| | S | Acc | P | R | F1 | S | Acc | P | R | F1 | S | Acc | P | R | F1 |
| RNP* | 10.0 | - | 32.4 | 18.6 | 23.6 | 10.0 | - | 44.8 | 32.4 | 37.6 | 10.0 | - | 24.6 | 23.5 | 24.0 |
| INVRAT* | 10.0 | - | 42.6 | 31.5 | 36.2 | 10.0 | - | 41.2 | 39.1 | 40.1 | 10.0 | - | 34.9 | 45.6 | 39.5 |
| Inter-RAT* | 11.7 | - | 66.0 | 46.5 | 54.6 | 11.7 | - | 55.4 | 47.5 | 51.1 | 12.6 | - | 34.6 | 48.2 | 40.2 |
| FR | 11.1 | 75.8 | 70.4 | 42.0 | 52.6 | 9.7 | 87.7 | 68.1 | 42.2 | 52.1 | 11.7 | 87.9 | 43.7 | 40.9 | 42.3 |
| MCD-KL | 9.5 | 79.7 | 94.2 | 48.4 | 63.9 | 9.9 | 87.5 | **84.6** | **53.9** | 65.8 | 9.4 | 87.3 | 60.9 | 47.1 | 53.1 |
| MCD-JS | 9.7 | 80.1 | **95.7** | **50.2** | **65.9** | 10.0 | 86.1 | 79.8 | 51.0 | 62.2 | 10.9 | 85.6 | **62.1** | **54.4** | **58.0** |
| RNP* | 20.0 | - | 39.4 | 44.9 | 42.0 | 20.0 | - | 37.5 | 51.9 | 43.5 | 20.0 | - | 21.6 | 38.9 | 27.8 |
| INVRAT* | 20.0 | - | 58.9 | 67.2 | 62.8 | 20.0 | - | 29.3 | 52.1 | 37.5 | 20.0 | - | 24.0 | 55.2 | 33.5 |
| Inter-RAT* | 21.7 | - | 62.0 | 76.7 | 68.6 | 20.4 | - | 44.2 | 65.4 | 52.8 | 20.8 | - | 26.3 | 59.1 | 36.4 |
| FR | 20.9 | 84.6 | 74.9 | 84.9 | 79.6 | 19.5 | 89.3 | 58.7 | 73.3 | 65.2 | 20.2 | 88.2 | 36.6 | 59.4 | 45.3 |
| MCD-KL | 20.0 | 85.5 | **79.3** | **85.5** | **82.3** | 19.3 | 88.4 | **65.8** | **81.4** | **72.8** | 19.6 | 87.7 | 41.3 | 65.0 | 50.5 |
| MCD-JS | 19.9 | 80.8 | 77.7 | 83.4 | 80.5 | 18.8 | 87.2 | 60.5 | 73.1 | 66.2 | 20.2 | 86.0 | **42.3** | **68.5** | **52.3** |
| RNP* | 30.0 | - | 24.2 | 41.2 | 30.5 | 30.0 | - | 27.1 | 55.7 | 36.4 | 30.0 | - | 15.4 | 42.2 | 22.6 |
| INVRAT* | 30.0 | - | 41.5 | 74.8 | 53.4 | 30.0 | - | 22.8 | 65.1 | 33.8 | 30.0 | - | 20.9 | 71.6 | 32.3 |
| Inter-RAT* | 30.5 | - | 48.1 | 82.7 | 60.8 | 29.4 | - | 37.9 | 72.0 | 49.6 | 30.4 | - | 21.8 | 66.1 | 32.8 |
| FR | 29.6 | 86.4 | 50.6 | 81.4 | 62.3 | 30.8 | 88.1 | 37.4 | 75.0 | 49.9 | 30.1 | 87.0 | 24.5 | 58.8 | 34.6 |
| MCD-KL | 29.7 | 86.7 | 59.6 | **95.6** | 73.4 | 29.6 | 90.2 | 46.1 | **87.5** | 60.4 | 29.4 | 87.0 | **30.5** | **72.4** | **42.9** |
| MCD-JS | 29.0 | 89.6 | **60.2** | 94.4 | **73.5** | 28.7 | 86.2 | **47.3** | 87.0 | **61.3** | 27.6 | 84.5 | 26.9 | 59.7 | 37.1 |

Since our MCD criterion (Equation 13) is not limited to a specific measurement of dependence, we also conduct experiments by replacing KL-divergence with JS-divergence. The results are in Table 10. With either KL-divergence or JS-divergence, our MCD criterion always beat all the MMI-based baselines, showing the effectiveness of MCD.

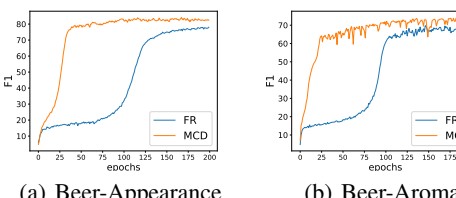

(a) Beer-Appearance  (b) Beer-Aroma

Figure 5: A comparison of convergence speed between our MCD and the latest MMI-based SOTA FR.

## A.6 Time efficiency

By avoiding many local optima, our MCD can converge much faster than MMI-based methods. Figure 5 shows a comparison of convergence speed between our MCD and the latest MMI-based SOTA FR on *Beer-Appearance* and *Beer-Aroma* with $S \approx 20$, where FR and MCD get the similar F1, and they use the same learning rate (0.0001) and batchsize (128).

## B Proofs

### B.1 Derivation of Equation 6

We use $X_S$ as an example, and the others are nothing different.

$$p(X_S = 1) = \sum_{U \in \{0,1\}} p(X_S = 1, U) = \sum_{U \in \{0,1\}} p(X_S = 1|U)p(U) = 0.9 * 0.5 + 0.1 * 0.5 = 0.5. \quad (24)$$

### B.2 Derivation of Equation 8 and 9

In Figure 2(a), we have $X_T \perp\!\!\!\perp X_S|U$ and $X_T \perp\!\!\!\perp Y_S|X_S$ (please refer to Appendix B.4). That is to say,

$$P(X_S|U, X_T) = P(X_S|U), \ P(Y_S|X_S, X_T) = P(Y_S|X_S). \quad (25)$$

Then we can easily get Equation 8:

$$\begin{aligned} p(X_S = 1|X_T = 1) &= \sum_{U \in \{0,1\}} p(X_S = 1, U|X_T = 1) \\ &= \sum_{U \in \{0,1\}} p(X_S = 1|U, X_T = 1)p(U|X_T = 1) \\ &= \sum_{U \in \{0,1\}} p(X_S = 1|U)p(U|X_T = 1). \end{aligned} \quad (26)$$

And Equation 9 is similar.

### B.3 The relation between entropy and cross-entropy

It is a basic idea in information theory that the entropy of a distribution $P$ is upper bounded by the cross entropy of using $Q$ to approximate it. For any two distribution $P$ and $Q$, we have

$$H_c(P,Q) = H(P) + D_{KL}(P\|Q) \geq H(P), \tag{27}$$

where the subscript $c$ in $H_c(P,Q)$ stands for cross-entropy.

We know that we get the minimum cross entropy when $Q$ is the same as $P$, i.e., $D_{KL}(P\|Q) = 0$. Which means

$$\min H_c(P,Q) = H(P). \tag{28}$$

### B.4 Conditional independence in a probabilistic graph

In the probabilistic graph depicted in Figure 6, we have that $A \perp\!\!\!\perp C|B$, $B \perp\!\!\!\perp D|C$, and $C \perp\!\!\!\perp E$ (but note that we do not have $C \perp\!\!\!\perp E|D$). This property is fundamental in probabilistic graphical models. The proof is straightforward, and we illustrate it using $A \perp\!\!\!\perp C|B$ as an example.

Based on the general principle of the chain rule, we can have

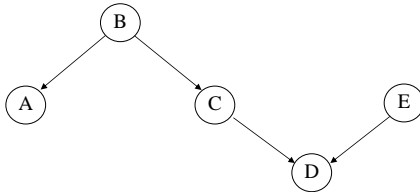

$$\begin{aligned} P(A,B,C) &= P(C|A,B)P(A,B) \\ &= P(C|A,B)P(A|B)P(B). \end{aligned} \tag{29}$$

Figure 6: A probabilistic graph that contains a fork, a chain, and a collider.

Based on the graph structure in Figure 6, we have

$$P(A,B,C) = P(B)P(A|B)P(C|B) \tag{30}$$

Combining Equation 29 and 30, we get

$$P(C|B) = P(C|A,B), \tag{31}$$

which means $A \perp\!\!\!\perp C|B$.

If you are seeking a more intuitive understanding of blocked path, please refer to a concept called "Bayes ball" (Jordan, 2003).

### B.5 Proof of Lemma 1

To prove this, we employ a proof by contradiction. We initially assume that $X_C$ is a variable in $X_R$ and $X_C \notin X_Z$. Given that $X_C \in X_R$, we deduce that $X_C$ exerts a direct causal influence on $Y$, i.e., there exists a path $X_C \to Y$:

$$X_C \in X_R \implies X_C \to Y.$$

Furthermore, since $X_C \notin X_Z$, we ascertain that $X_C \in X_{-Z}$. Consequently, we understand that $X_{-Z}$ and $Y$ are not d-separated by $X_Z$ due to an unblocked path $X_C \to Y$.

With this, we complete the proof of Lemma 1.

### B.6 Failure cases of Assumption 1

As far as we know, most of the real-world datasets are built in a collecting-annotating form. In such a form, $Y$ is given according to $X$, and the annotators won't edit $X$ after giving $Y$. So, Assumption 1 holds. But there are also some cases that might break Assumption 1. One is the synthetic data like ColorMinist. In ColorMinist, a human first annotates an image, and then edits the image again according to the assigned label. Another scenario is the collection of time series data, where annotators label the data based on existing information and then adjust the data collection method according to the previous labels. This creates a cyclic causal graph. However, in the literature of causal inference, most researchers only consider acyclic graphs. We note that in cases where Assumption 1 doesn't hold, we still have Lemma 1, i.e., D-separation severs as a sufficient condition for selecting causal rationales. When Assumption 1 holds, it becomes a necessary and sufficient condition.

### B.7 Proof of Lemma 2

To prove it, we employ a proof by contradiction. We first assume a variable $X_C \in X_{-Z}$, and $X_C$ is associated with $Y$ conditioned on $X_Z$. To achieve the association, there must be a path in either of the following two forms. The first form is

$$X_C \cdots o \to Y, \quad s.t. \ o \notin X_Z, \tag{32}$$

where "$\cdots$" denotes some arbitrary arrows and nodes, and $o$ is a intermediate node. $o \notin X_Z$ is from that if $o \in X_Z$, the path will be blocked by $X_Z$.

Since $o$ is a direct cause of $Y$, we have $o \in X_R$. Since $X_R \subset X_Z$, but we have $o \notin X_Z$, so this form of paths do not exist.

The second form is

$$X_C \cdots \to o \leftarrow o_1 \leftarrow \cdots o_n \leftarrow Y, \quad s.t. \ o \in X_Z, \tag{33}$$

where $o_1 \cdots o_n$ are some nodes connected by left arrows, we do not discuss these nodes since discussing $o$ is enough for our proof.

This path is unblocked through a collider. Note that the way to unblock a collider path is to condition on it, so we need to have $o \in X_Z$. However, in this case, $Y$ has a causal effect on $o$, which breaks Assumption 1. So, this form of paths do not exist as well.

As a result, there is no variable in $X_{-Z}$ can be associated with $Y$ conditioned on $X_Z$. The proof of Lemma 2 is completed.