# OpenReview forum: "D-Separation for Causal Self-Explanation"
_NeurIPS.cc/2023/Conference — NeurIPS 2023 poster_

### Official Review · Reviewer_bHMi · 2023-07-03

**Soundness:** 3 good
**Presentation:** 3 good
**Contribution:** 2 fair
**Rating:** 3
**Confidence:** 5

**Summary:**

In this paper, the authors focus on self-explaining rationalization and they propose a Minimum Conditional Dependence (MCD) criterion to select causal rationales from a causal perspective.

**Strengths:**

1. Rationalization is a worthwhile direction to explore in interpretable research.
2. The paper is well written.

**Weaknesses:**

1.This paper lacks novelty. First of all, the causalities in rationalization has been widely discussed[1][2][3]. Besides, the method proposed by the authors is very similar to the previously proposed method[4][5]. Authors do not compare them as baselines and not cite these research paper.

2. The datasets in this paper are limited and I would expect to see more datasets such as ERASER[6] that have been widely used for rationalization[7][8][9].
Furthermore, both BeerAdvocate and HotelReview seem to focus on the binary classification, have the authors studied the effectiveness of MCD on multi-classification tasks, such as DARE[10] and InfoCAL[5] do?

3. Can the authors provide an example to show that the rationales extracted by MCD extraction is not affected by spurious correlations in the data.

4. The recent work is insufficient where much of the recent work on the rationalization is not discussed, such as [3][4][5][6][8][10].

References
[1] Invariant rationalization.
[2] Interventional rationalization.
[3] Discovering Invariant Rationales for Graph Neural Networks.
[4] How do Decisions Emerge across Layers in Neural Models? Interpretation with Differentiable Masking
[5] Learning from the Best: Rationalizing Prediction by Adversarial Information Calibration
[6] ERASER: A benchmark to evaluate rationalized NLP models.
[7] An information bottleneck approach for controlling conciseness in rationale extraction.
[8] Unifying model explainability and robustness for joint text classiﬁcation and rationale extraction.
[9] UNIREX: A Unified Learning Framework for Language Model Rationale Extraction
[10] DARE: Disentanglement-augmented rationale extraction


**Questions:**

1.Why do the authors use different baselines in BeerAdvocate and HotelReview?

2.The authors' discussion of rationalization+LLMs is insufficient, I would like to see more of the authors' views on rationalization+LLMs.

3. The authors' experiments focus on the text classification task, would MCD be equally effective on other nlp tasks, such as text generation?



**Limitations:**

See Weaknesses for details.

---

> ### Author Rebuttal · Authors · 2023-08-09
>
> Thank you very much for your valuable comments and suggestions.
>
> **A1 (Novelty):**
> Regarding the comparison with [1,2,3], we agree that the causality in rationalization is not a new topic. However, our research differs significantly from [1,2,3]. The limitaions of [1,2] ([3] is similar to [1]) have been discussed in details in Sec.1 and App.C. For the methodology, our approach is not inspired by either IRM or backdoor adjustment. The analyses on rationale causality are also very different from [1,2]. We also  provide a very different theoretical perspective linking causality to our MCD criterion, which is model-agnostic and could enhance other methods. Furthermore, our performance significantly outperforms [1] and [2] in terms of F1 score.
>
> Compared to [4], our research targets a different problem, guided by unique motivations and novel theoretical insights. While [4] focuses on post-hoc explanation, our work is centered on self-explanation literature, leading to different challenges (faithfulness for post-hoc explanation (page 4 in [4]) and plausibility for self-explanation (L27-65)) and targets. Besides, [4] doesn't touch any analysis about the causality. The focus of [4] is developing differentiable masking. Our main contributions, the analyses on rationale causality, diverge from [4].
>
> [5] may appears similar as it uses a discriminator to make the guider predictor and the original predictor encode indistinguishable information; however, our goals and theoretical insights differ significantly from [5]. For example, [5] cannot guarantee that causal rationales will get lower $l_d$ (the discrimination loss in [5]) than spurious correlations. Also, [5] doesn't align closely with our research problem (spurious correlations and degeneration), and its code isn't accessible.
>
> Could you please provide more details about the similarity so that we can better address your concerns? We appreciate your suggestion and will discuss [3,4,5] in Sec.2.
>
> We respectfully highlight that our work is not only about proposing a new method, but also about linking two key problems to the flexible MCD criterion. As far as we know, the degeneration has never been considered as a non-causality problem, even in [1,2].
>
> **A2 (datasets):** Our experiments focus on testing the efficacy of our MCD in selecting causal rationales, which calls for specific dataset properties. First, the main challenge should be to distinguish between association and causation rather than other problems, which requires there to be some spurious correlations in the dataset. Multi-aspect classification datasets satisfy this property well, since there are usually correlations between different aspects. Other datasets in ERASER don't satisfy this property. Second, there should be human-annotated causal rationales in the test set to measure whether the selected rationales are real causal rationales or just associated with the label. The LJP dataset used in [5,10] doesn't satisfy this property. [5,10] only evaluate the prediction performance on this dataset, which is only association, not causation.
>
> **A3 (examples):** Thank you for your suggestion. It was an oversight on our part to leave that out. We have added some examples to Fig.1 and 2 of the **rebuttal.pdf**.
>
> **A4 (recent references):** We appreciate your suggestion and will include them in Sec.2. Given the vastness of the rationalization field, it's hard to cover all new papers in a non-survey paper. However, we've exhaustively discussed all important papers that pertain to our key research problems, i.e., feature correlation and degeneration.
>
> **A5 (different baselines for Beer and Hotel):** Thank you for your question. We have now reimplemented Inter_RAT on Hotel. The results are in Table 1 of the **rebuttal.pdf**. Since INVRAT doesn't provide runable code and the details of how to create different environments is not very clear, we fail to reimplement INVRAT on other datasets. Another reason we didn't include INVRAT and Inter_RAT in Table 3 is that FR (published in late 2022) has been shown to outperform INVRAT and Inter_RAT a lot on BeerAdvocate (which is the main dataset used in INVRAT and Inter_RAT), so we thought comparing MCD to FR was somewhat enough.
>
>
> **A6 (LLMs):** Here is a brief discussion of LLMs+XAI. With the great success of LLMs, a new research line for XAI is chain-of-thought. By generating (as opposed to selecting) intermediate reasoning steps before inferring the answer, the reasoning steps can be seen as a kind of explanation. However, LLMs sometimes exhibit unpredictable failure modes [B] or hallucinatory reasoning [C], making this kind of generative explanation not trustworthy enough in some high-stakes scenarios. Also, some recent research finds that LLMs are not good at extractive tasks [D-F].
> We appreciate your suggestion, but delving further into the challenges of applying LLMs is somewhat beyond the scope of this paper. We will add it to the limitations and leave it as future work.
>
> **A7 (would MCD be equally effective on other nlp tasks?):**  It depends on the problem definition. As you know, different tasks can have different causal structures. We hope that the motivational process of MCD can inspire others to propose methods that work in other fields. However, we don't claim that our method is directly applicable to any tasks without modification.
>
> [A] ZIN: When and How to Learn Invariance Without Environment Partition? NeurIPS 2022.
> [B] Causal reasoning and large language models: Opening a new frontier for causality. arXiv:2305.
> [C] Survey of hallucination in natural language generation. ACM Computing Surveys, 2023.
> [D] Is chatgpt a general-purpose natural language processing task solver? arXiv:2302.
> [E] Evaluating chatgpt’s information extraction capabilities: An assessment of performance, explainability, calibration, and faithfulness. arXiv:2304.
> [F] A comprehensive capability analysis of gpt-3 and gpt-3.5 series models. arXiv:2303.

---

> > ### Author Response · Authors · 2023-08-17
> > **Further clarifications on the novelty and datasets.**
> >
> > **A1:** Dear reviewer, I'd like to further elaborate on the distinct novelty of our work in comparison to [4].
> >
> > Before we begin, we'd like to emphasize that our primary contribution lies in the novel theoretical insights regarding rationale causality and the introduction of the MCD criterion as a replacement for MMI, rather than in a specific model (L105-109).
> >
> > Below are some distinctions from the perspective of the proposed model:
> >
> > While [4] focuses on post-hoc explanation, our study concentrates on self-rationalization. These are two distinct research areas.  The primary challenge in post-hoc explanation is ensuring faithfulness (see L27-33 of our submission and Section 3 of [4]), which means the explanation should correspond closely with the model's prediction, $\hat{Y}$. This intuitively leads to aligning $P(\hat{Y}|X)$ with $P(\hat{Y}|X_Z)$. But in our paper, we are extracting rationales that are causal to the human-annotated **gold label $Y$**. Note that from a theoretical standpoint, even subtle differences can lead to very different results.
> >
> > Very different from [4], the goal of our Equation 16 is designed to align  $P({Y}|X)$ and $P({Y}|X_Z)$ (L252).
> > This necessitates minimizing the cross-entropies to approximate both $P({Y}|X)$ and $P({Y}|X_Z)$ via $P(\hat{Y}|X)$ and $P(\hat{Y}|X_Z)$, respectively (L253-257). The above motivations are all based on our causality analysis, which is central to our research.
> > [4] doesn't touch on causality analysis, and as a result there is no approximation for $P({Y}|X_Z)$ in [4].
> >
> >  Here are the ablation experiments that remove the approximation for $P({Y}|X_Z)$:
> > | Appearance | S | Acc | P | R | F1 || S | Acc | P | R | F1 ||S | Acc | P | R | F1 |
> > |:---:|---|---|---|---|---|-|---|---|---|---|---|-|--|---|---|---|---|
> > | MCD | 9.5 | 81.5 | 94.2 | 48.4 | **63.9** || 20.0 | 85.5 | 79.3 | 85.5 | **82.3** || 29.7| 86.7 | 59.6 | 95.6 | **73.4** |
> > | w/o $H_c(Y,\hat{Y}_z\|X_Z)$ | 9.7 | 72.5 | 85.1 | 44.6 |58.5  || 20.0 | 75.5 | 74.6 | 80.7 |77.5  || 32.4 | 81.2 | 54.6 | 95.5 |69.5  |
> >
> > | Aroma | S | Acc | P | R | F1 || S | Acc | P | R | F1 || S | Acc | P | R | F1 |
> > |:---:|---|---|---|---|---|-|---|---|---|---|---|-|---|---|---|---|---|
> > | MCD | 9.9 | 87.5 | 84.6 | 53.9 | **65.8** || 19.3 | 88.4 | 65.8 | 81.4 | **72.8** || 29.6 | 90.2 | 46.1 | 87.5 | **60.4** |
> > | w/o $H_c(Y,\hat{Y}_z\|X_Z)$ | 11.4 | 86.3 | 70.2 | 51.6 |59.4  || 21.4 | 84.3 | 58.8 | 80.6 |68.0  || 30.4 | 88.4 | 40.4 | 78.8 |53.4  |
> >
> > | Palate | S | Acc | P | R | F1 || S | Acc | P | R | F1 || S | Acc | P | R | F1 |
> > |:---:|---|---|---|---|---|-|---|---|---|---|---|-|---|---|---|---|---|
> > | MCD | 9.4 | 87.3 | 60.9 | 47.1 | **53.1** || 19.6 | 87.7 | 41.3 | 65.0 | **50.5** || 29.4 | 87.0 | 30.5 | 72.4 | **42.9** |
> > | w/o $H_c(Y,\hat{Y}_z\|X_Z)$ | 10.6 | 83.7 | 53.0 | 45.2 |48.8  || 20.5 | 85.2 | 37.3 | 61.4 |46.4  || 31.5 | 86.5 |25.4 | 64.4 |36.4  |
> >
> > We see that when we remove the approximation, there is a significant drop in the F1-score, demonstrating the importance of our novel theoretical analyses.
> >
> > We are grateful for your time and expertise in reviewing our paper. To better address your feedback, could you elaborate on the similarity concerns you raised?
> >
> > **A2:** Here are some further clarifications about the ERASER datasets.
> >
> > We think there is a misunderstanding. While ERASER is indeed a valuable resource, it may not always be the best benchmark for every research problem.
> >
> > The primary strength of ERASER datasets is the inclusion of human-annotated rationales in the training set, making them particularly beneficial for supervised rationale extraction. To the best of our knowledge, the majority of methods that leverage ERASER datasets, including those you've mentioned ([7,8,9]), all require human-annotated rationales for training. In contrast, methods that focus on unsupervised rationale extraction, such as [1,2,5,10] that you mention, tend not to employ ERASER. Currently, the Beer dataset remains a predominant choice in this area. We opted for the Beer and Hotel datasets, aligning with the recent and strong baseline FR from NeurIPS 2022.

---

> > ### Comment · Reviewer_bHMi · 2023-08-20
> > **Response to rebuttal**
> >
> > Thanks for the response. But I still have the following questions:
> >
> > **about A2**
> > I do not agree with the authors' explanation for not using the ERASER dataset. Since MCD should eventually be applied in practice, it should be validated on more datasets to verify the effectiveness of MCD. Moreover, the datasets in ERASER all contain true rationale labels and can be considered as the causal rationale.
> >
> > **about A4**
> > I don't agree with the authors' statement "Given the vastness of the rationalization field, it's hard to cover all new papers in a non-survey paper". Some of the relevant work I mentioned in the review comments was done before 2022. For NeurIPS 2023, they are not new papers.

---

> > > ### Author Response · Authors · 2023-08-21
> > > **We are grateful for your feedback.**
> > >
> > > We are grateful for your feedback. Here are some further clarifications.
> > >
> > > **A1 (further clarification on dataset selection):** We agree that the application is important. But as a research paper, the most important role of the experiments is to verify the theoretical claims, rather than to achieve the engineering SOTA. This is because different applications face different challenges, and different methods are aimed at solving different problems. We can choose different methods for different applications.
> > > As implied in the previous rebuttal, the most appropriate dataset for verifying the ability to select causal rationales is still the BeerAdvocate, and the methods that are most relevant to our research (INVRAT, Inter_RAT, and FR) all use BeerAdvocate as their main experiment. Also, the datasets we chose are just the same as the strongest baseline FR (NeurIPS 2022).
> > >
> > > There are two lines of research in rationalization: supervised rationalization and unsupervised rationalization, where "supervised" means that human-annotated rationales are required for training. As implied in the previous rebuttal, the primary advantage of the ERASER datasets is the inclusion of human-annotated rationales in the training set, making them particularly useful for supervised rationale extraction. To the best of our knowledge, the majority of methods that leverage the ERASER datasets, including those you've mentioned ([7,8,9]), are all supervised methods. However, our research is unsupervised rationalization, and the datasets we chose are those widely used by other unsupervised methods. Our datasets align with two recent papers DMR (AAAI2021) and FR (NeurIPS 2022). And we recently found that a recent ( June 25, 2023) published paper CR[A] also uses BeerAdvocate as the main experiment and also uses HotelReviews as a supplement, without using ERASER, which invalidates the usefulness of the BeerAdvocate and HotelReviews datasets we used.
> > >
> > > **A2 (new references):** Thank you for your suggestion. We agree that some of the papers in the field of rationalization were not covered in the original submission, and we will discuss them in Sec.2 in the next version. What I mean by "it's hard to cover all new papers in a non-survey paper" is that we only discussed the papers that study similar research topics (i.e., feature correlation and degeneration) with us in detail, and ignored those sub-relevant papers, due to the page limit. We appreciate these valuable references and we will follow up on your suggestion to make a more comprehensive survey.
> > >
> > > We appreciate your continued dedication and effort in evaluating our manuscript.
> > >
> > > [A] Towards Trustworthy Explanation: On Causal Rationalization. ICML 2023.

---

> > > > ### Comment · Reviewer_bHMi · 2023-08-21
> > > > **Thank you for the author's response.**
> > > >
> > > > Thank you for the author's response. However, the author's reply has not addressed my concerns, including the novelty and comprehensiveness of the experiments.
> > > >
> > > > For the ERASER dataset, both the test set and training set have manually annotated rationales, while the beer dataset only has manually annotated rationales in the test set. I acknowledge the usefulness of the beer dataset, but I still insist that the author should conduct experiments on the ERASER benchmark dataset.
> > > >
> > > > Furthermore, the author's understanding of [7] is incorrect. IB is still an unsupervised rationalization, and it is highly scalable, capable of extending to supervised rationalization. However, its core part (Figure 2 in [7]) is an unsupervised rationalization. I hope the author can carefully read the work I mentioned and improve the related work section.

---

> > > > > ### Author Response · Authors · 2023-08-21
> > > > > **Thank you for the timely feedback. And could you please provide more details about the novelty concerns?**
> > > > >
> > > > > We appreciate your suggestions and are already conducting experiments on one of the ERASER datasets (i.e., Movie), which will be completed soon. Could you elaborate more on your concerns about novelty? If you could pinpoint the specific similarities between our paper and others, we believe we can address your questions more effectively.
> > > > >
> > > > > We are aware that the framework in [7] can be adapted for both supervised and unsupervised settings. However, in the experiments of [7], they employed a semi-supervised setup where 25% of the training data requires human-annotated rationales. This necessitates the use of the ERASER dataset, which is why we categorize it under supervised methods. We appreciate you pointing this out and will read the new references carefully to improve the related work section.

---

> > > > > > ### Author Response · Authors · 2023-08-21
> > > > > > **Additional experiments on ERASER.**
> > > > > >
> > > > > > We have now performed additional experiments on one of the ERASER datasets: **Movie**. And here are the results:
> > > > > >
> > > > > > | Method | P | R | F1 |
> > > > > > |:---:|---|---|---|
> > > > > > | Inter_RAT\* | 35.7 | 35.8 | 35.7 |
> > > > > > | FR | 36.0 | 36.4 | 36.2 |
> > > > > > | MCD (Ours) | 37.6 | 38.9 | **38.2** |
> > > > > >
> > > > > > Note that INVRAT relies on the labels of other aspects to create different environments to remove feature correlations, so we cannot compare with INVRAT on this non-multi-aspect classification dataset. However, we can see that INVRAT performs worse than Inter_RAT and FR on the Beer dataset. So on this dataset, we only compare with Inter_RAT and FR.
> > > > > >
> > > > > > "\*": The results of Inter_RAT are obtained from its original paper. We follow Inter_RAT to set the rationale sparsity to be about 20%. Since there are not many feature correlations in this dataset, we don't get as much improvement as in the Beer and Hotel datasets. But mask correlations can still occur in this dataset, so our MCD still gets considerable improvements over MMI-based methods ($\frac{38.2-36.2}{36.2}\approx$5.5%).
> > > > > >
> > > > > >
> > > > > > Here is a list of the papers that are close to our research topics (i.e., feature correlation or degeneration) and their corresponding datasets:
> > > > > > | Method | Venue | Datasets |
> > > > > > |:---:|---|---|
> > > > > > | INVRAT  | ICML 2020 | Beer+IMDB |
> > > > > > | Inter_RAT  | unpublished| Beer+Movie+LJP |
> > > > > > | DMR  | AAAI 2021 | Beer+Hotel |
> > > > > > | A2R  | NeurIPS 2021 | Beer+Movie |
> > > > > > | FR  | NeurIPS 2022 | Beer+Hotel |
> > > > > > | CR [A] | ICML 2023 | Beer+Hotel+GA |
> > > > > > | MGR [B] | ACL 2023 | Beer+Hotel |
> > > > > > | DR [C] | SIGKDD 2023 | Beer+Hotel |
> > > > > >
> > > > > > Notes: LJP and GA datasets have no human-annotated rationales for testing.
> > > > > > Our choice of datasets (Beer and Hotel) seems reasonable, as this choice is also widely used by other methods.
> > > > > >
> > > > > > Thank you very much for taking the time to review our paper! With best wishes to you and yours!
> > > > > >
> > > > > > [A] Towards Trustworthy Explanation: On Causal Rationalization. ICML 2023.
> > > > > > [B] MGR: Multi-generator Based Rationalization. ACL 2023.
> > > > > > [C] Decoupled Rationalization with Asymmetric Learning Rates: A Flexible Lipschitz Restraint. SIGKDD 2023.

---

### Official Review · Reviewer_kLrd · 2023-07-05

**Soundness:** 2 fair
**Presentation:** 3 good
**Contribution:** 2 fair
**Rating:** 5
**Confidence:** 4

**Summary:**

This paper studies selective rationalization. Many methods use the maximum mutual information (MMI) criterion to find the most indicative rationale to explain a target label. As it has been shown in the past, this criterion is, by design, sensitive to spurious correlation. This paper proposes a novel criterion instead of "fixing" MMI, called Minimum Conditional Dependence (MCD). The authors identify two stages from which spurious correlations may come from and propose a causal framework to circumvent them.

Method
The paper assumes that the beer dataset has been generated on 3 aspects: Appearance, Taste, and Smell. This is wrong since the dataset has 5 aspects in total (palate and overall). Nevertheless, the motivating example is convincing. Leveraging the concept of d-separation is novel. However, it is unclear how dissimilar is the proposed approach w.r.t. [1,2] for example (L240 and L130 don't discuss really the differences) A downside of the approach is the need to train one model per aspect. It would interesting to discuss about extending the proposed model.

In the experiment section, I would appreciate having the results for the other aspects, and potentially, the original beer dataset to understand the effect of MCD (while not many papers utilize the original beer dataset, I do think it's important). The number of baselines is not consistent through the experiments: T3-T4 would require more models to assess the superiority of MCD.
Finally, I am surprised that MCD performs on average better rationalization performance but worst predictive performance, which defeats the purpose of "rationalization": explaining the output of a model. Why is there such a gap?

The related work is quite complete. Nevertheless, I have identified few missing citations [3-5].

1 Chang et al. 2021, Invariant Rationalization (ICML)
2 Yue et al. 2022, Interventional Rationalization
2 Chang et al. 2019, A Game Theoretic Approach to Class-wise Selective Rationalization (NeurIPS)
3 Antognini et al. 2021, Multi-Dimensional Explanation of Target Variables from Documents (AAAI)
4 Antognini and Faltings 2021, Rationalization through Concepts (ACL)

**Strengths:**

- The causal framework is novel, and using d-separation is a nice replacement to circumvent MMI's flaws
- MCD obtains better rationalization performance
- The paper is well written and motivated.

**Weaknesses:**

- It is unclear how dissimilar is the proposed approach w.r.t. [1,2] for example (L240 and L130 don't discuss really the differences)
- The proposed model only works for 1 aspect.
- MCD obtains worst predictive performance, which defeats the purpose of "rationalization": explaining the output of a model.

**Questions:**

- Why MCD underperforms in terms of predictive performance?
- How could we extend MCD to multi-dimensional rationales?
- Could you report results of MCD with BERT for the other experiments since it provides better performance?

**Limitations:**

The authors do not discuss about the limitations and potential negative societal impact of their work.

---

> ### Author Rebuttal · Authors · 2023-08-09
>
> We greatly appreciate your detailed review and constructive feedback on our paper!
>
> **A1 (regarding [1,2]):** Our key differences with [1,2] are discussed in L66-85 of Sec.1 and App.C, where we delve into their limitations and theoretical flaws. [1] employs IRM and [2] uses backdoor adjustment to add regularizers to the MMI loss. Our approach is not inspired by either IRM or backdoor adjustment. We bring new theoretical insights linking causality to our model-agnostic MCD criterion, potentially benefiting other methods. Moreover, our results surpass those of [1] and [2] by substantial margins, with improvements of 19.1 and 13.1, respectively, averaging over nine settings in Table 2's F1 scores.
>
> **A2 (only works for 1 aspect):** Could you please provide a clearer delineation for the term "1 aspect"? We see that MCD outperforms all baselines in terms of rationale quality across all aspects on the standard BeerAdvocate and HotelReview benchmarks in Table 2-3.
>
> **A3 (worst predictive performance):** Our analysis suggests otherwise, could you elaborate on your concerns? In the nine settings of BeerAdvocate, our MCD secured the highest accuracy in four cases, was only marginally behind the best FR in another four, and matched FR in the remaining one. Moreover, our results consistently outperformed INVRAT. On the standard Beer and Hotel datasets, our accuracy drop, compared to the best method, was at most 0.9%. Overall, when averaged across aspects or sparsities, our performance matches or exceeds the best baseline FR.
>
> The accuracy of FR and MCD on BeerAdvocate (Table 2):
> ||S|Apearance|Aroma|Palate|Average
> :-:|:-:|:-:|:-:|:-:|:-:
> FR|10|75.8|87.7|87.9|83.8
> MCD|10|81.5|87.5|87.3|**85.4**
> :-:|:-:|:-:|:-:|:-:|:-:
> FR|20|84.6|89.3|88.2|**87.4**
> MCD|20|85.5|88.4|87.7|87.2
> :-:|:-:|:-:|:-:|:-:|:-:
> FR|30|86.4|88.1|87.0|87.2
> MCD|30|86.7|90.2|87.0|**87.9**
> :-:|:-:|:-:|:-:|:-:|:-:
> FR|average|82.3|88.4|**87.7**|86.1
> MCD|average|**84.6**|**88.7**|87.3|**86.9**
>
> **A4 (Why lower predictive performance):** I guess you are referring Table 4 to say that MCD gets the worst predictive performance. The synthetic experiments of Table 4 are crafted to demonstrate MCD's efficacy against degeneration. Note that degeneration might not adversely affect predictive accuracy; sometimes it can even improve it. In Table 4, as the rationale quality decreases with increasing skew, the accuracy of RNP conversely rises (in the settings of Table 4, the explainer is specially initialized to select trivial patterns to indicate the label. If the predictor fits such patterns, the eplainer and predictor can collude to get high accuracy. But if the predictor fits the true semantics, the accuracy will not be as high). However, such elevated accuracy might be unstable and drop with shifts in trivial patterns. The accuracy of our MCD, though lower than that of RNP and FR, is more aligned with the rationale quality, indicating less degeneration.
>
> Notably, a recent study claims that relying on causal features doesn't always guarantee optimal accuracy [A].
>
> **A5 (extend MCD to multi-dimensional rationales):** While it's typical in current methods to train a model for each aspect, we agree that a consolidated approach is desirable. Research in this direction, such as the valuable references you provided, offers promising techniques. We're contemplating leveraging their insights to refine MCD for multi-dimensional rationale extraction. However, this endeavor remains a future aspiration, and is somewhat beyond the scope of this paper.
>
> **A6 (more results with BERT):** We are sorry, but due to limited GPU resources, only six of the nine experiments from Table 2 have been completed in time, and the results are shown in Table 3 of the rebuttal.pdf.
> Most prior methods underperform when using BERT. The fine-tuning intricacies of BERT make it difficult to verify specific reasons for performance improvements. Thus, we primarily employed GRUs to validate MCD's efficacy. Also, due to resource limitations, we weren't able to conduct all experiments with BERT. We plan to address this in future work and will note this limitation in our paper. Thank you for your suggestion.
>
> **A7 (original beer):** The experiments in Table 2 are run on the non-decorrelated dataset. We now report the results for the taste aspect (since all other aspects can serve as a causal part of the overall label, we don't include the overall aspect) in Table 2 of the **rebuttal.pdf**. (We do not reimplement INVRAT on this aspect, and the reasons are in **A9**).
>
> **A8 (references):** Thank you for providing us such valuable references. We find they cover very interesting topics that we hadn't considered.  We will discuss them in Section 2 and consider drawing inspiration from them to expand our approach to multi-dimensional rationales in the future.
>
> **A9 (baselines in T3-4):** Thank for your suggestion. We have now reimplemented Inter_RAT and added it to Table 3. The results are in Table 1 of the **rebuttal.pdf**. Since INVRAT doesn't provide runable code and the details of how to create different environments are not very clear, we fail to reimplement INVRAT. Another reason we didn't include INVRAT and Inter_RAT in Table 3 is that FR has been shown to outperform INVRAT and Inter_RAT a lot on BeerAdvocate (which is the main dataset used in INVRAT and Inter_RAT), so we thought comparing MCD to FR was somewhat enough.
>
> Table 4 works more as an ablation study (where feature correlations are ablated) to verify the effectiveness in addressing degeneration. FR is specifically designed to address degeneration and has achieved SOTA results in that direction. We compare our MCD with vanilla RNP to validate its effectiveness and with FR to assess its competitiveness. The competitiveness of our MCD in real-world scenarios is primarily validated by Table 2-3, rather than Table 4.
>
> [A] Spuriosity Didn't Kill the Classifier: Using Invariant Predictions to Harness Spurious Features. arXiv:2307.

---

> > ### Author Response · Authors · 2023-08-21
> >
> > Thank you very much for taking the time to review our paper! With best wishes to you and yours!

---

### Official Review · Reviewer_Rbzy · 2023-07-06

**Soundness:** 3 good
**Presentation:** 3 good
**Contribution:** 3 good
**Rating:** 7
**Confidence:** 4

**Summary:**


The maximum mutual information criterion is commonly used for rationalization, but it uncovers associations rather than causal relationships. The authors propose to identify non-causal features that are independent of the labels given the causal features and the ‘minimum conditional dependence’ criterion, which does not require prior expert knowledge. Experiments are focused on the popular BeerAdvocate and HotelReviews datasets, though mostly on the former, and obtain very competitive results against the other, well-chosen baselines.

**Strengths:**

- The treatment of this problem using causal terminology is appropriate and interesting, and attempts were made to formalize some of these concepts (in Sec 4.2), with only some challenges, listed below.

- Appropriate baselines and datasets are provided in the experiments, as per other related (uncited) work. Moreover, attempting to overcome limitations of ‘some of the baseline methods’ (L290) by establishing consistent ‘settings’ is good, although what those settings are (beyond using GloVE) should be included.

- The rates of improvement in Table 2 are impressive. In some cases, tests of statistical significance would be useful to include, at least against FR, though apparently not necessary.

**Weaknesses:**

- The treatment of related works is extremely superficial in Section 2. Approximately half of the section on rationalization is merely a list of papers barely described by their topics and summarized in whole as being ‘orthogonal’. To some extent, a literature survey was better covered in Section 1,

- There are various minor issues including:
	- atypical English (L2 “pieces of their inputting texts”; L48 “is easily to be affected”; L56 “[lowercase] comments regarding”; L188 “a image”)
	- very small text (Fig 1)
	- incomplete citations (e.g., Yue et al (2023), L514.)

**Questions:**

- This seems like a companion piece to “MGR: Multi-generator Based Rationalization” by Liu et al (2023), as much of the text is identical and they also deal with spurious correlation and degeneration, but the focus is on the MCD criterion rather than using multiple generators. What other differences can easily be described in your paper, if this paper were cited?

- Is your ‘derivation’ in appendix B1 really an example of Bayes’ theorem, or just an application of marginalization and the chain rule?

- Assumption 1 (L223) should be better explained in Appendix B, as the ‘temporal sequence’ explanation provided in the paper is insufficient, especially if it refers to some aspect of dataset acquisition or prediction rather than the actual state of the world. Can you expand on this?

- the FR method (L274) is claimed to be the SotA, but even the paper by Liu et al (2022) to which it refers does not make a compelling case for it being the true SotA — how is this claim proven conclusively?

- Given the new methodological approach, it would be interesting to include some study of the required computational resources in the main body of the text, beyond the brief mention of RTX3090 and appendix A.5’s epochs, including possible ablations. Would that be possible?

- Can you add an explanation (or rationale) for why fine-tuning BERT would be ‘challenging’ (L299) beyond just providing citations and some indication of potential overfitting (section beginning L331)?

**Limitations:**

- The last paragraph of Sec 6 touches briefly on the limitations. It is suggested that not computing the ‘precise values’ of causal effects would be a limitation, although it is not clear why that would be. Various other forms of limitation, including the relatively restricted datasets (in terms of task, scope, and number) used in the experiments or the focus on text, could also be addressed, for example.

---

> ### Author Rebuttal · Authors · 2023-08-09
>
> We sincerely thank you for dedicating your time and expertise to review our paper. Your insightful comments and suggestions are highly valued and appreciated.
>
> **A1 (related work):** Thank you for pointing out this issue. Due to page limitations, we foucsed mainly on papers closely tied to our work, especially those that investigate spurious correlations. However, we agree with you that the first paragraph of Section 2 was a bit too brief. We will expand this content in the next version to provide a more complete understanding for new readers outside this field.
>
> **A2 (minor issues):** Thank you for your observation. We'll ensure to thoroughly revise and proofread the document in our next iteration to correct any minor issues. And the settings (Strengths 2, L290) include word embedding (GloVE), networks (GRUs), and similar sparsities (10%, 20%, 30%).
>
> **A3 (regarding MGR):**
> - First, the problem identification. Although MGR can address spurious correlation and degeneration simultaneously with only one model, in the context of MGR, spurious correlation and degeneration are considered as two separate problems and treated with distinct theoretical analyses. However, our MCD is the first to unify these two problems into one: they both arise from the non-causality of MMI. As a result, we can better understand these two problems with a unified theoretical insight.
>
> - Second, the potential implications. The core idea behind MGR to mitigate the influence of spurious correlations is the central limit theorem, which involves using more generators to reduce the probability of missing causal rationales. However, the model's lack of flexibility makes it less compatible with other rationalization variants. Our MCD, an optimization criterion, is model-agnostic and more versatile. Besides, we hope that the motivational process of our MCD can inspire research in other causal discovery fields, such as causal representation learning.
>
> - Third, the model complexity. MGR relies on the central limit theorem and involves multiple generators, which requires more computational resources than other methods.
>
> **A4 (derivation in App.B1):** It's just an application of marginalization and the chain rule. Thank you for your careful observation, and we will change the statement accordingly.
>
> **A5 (regarding Assumption 1):** As far as we know, most of the real-world datasets are built in a collecting-annotating form. In such a form, $Y$ is given according to $X$, and the annotators won't edit $X$ after giving $Y$. So, Assumption 1 holds.
> We really appreciate your insight very much, which inspires us to think of some cases that might break Assumption 1. One is the synthetic data like ColorMinist. In ColorMinist, a human first annotates an image, and then edits the image again according to the assigned label. Another scenario is the collection of time series data, where annotators label the data based on existing information and then adjust the data collection method according to the previous labels. This creates a cyclic causal graph. However, in the literature of causal inference, most researchers only consider acyclic graphs. Nevertheless, we greatly appreciate your insight a lot and will add the discussion to the Limitations section in the next version.
> We note that in cases where Assumption 1 doesn't hold, we still have Lemma 1, i.e., D-separation severs as a sufficient condition for selecting causal rationales. When assumption 1 holds, it becomes a necessary and sufficient condition.
>
> **A6 (FR SOTA):** FR was introduced in late 2022 and showed superior performance over recent baselines DMR (AAAI 2021) and A2R (NeurIPS 2021) on both the Hotel and Beer datasets. Our own tests also showed that FR outperformed INVRAT and Inter_RAT in most scenarios on the correlated Beer dataset (Table 2). Therefore, we suggested it as a current SOTA. We acknowledge this might not be entirely accurate, and we'll modify this statement in our revised version.
>
> **A7 (computational resources):** Thank you for your valuable suggestion. Taking the appearance aspect of BeerAdvocate as an example, the computational resources costed by different methods are:
> |  |batchsize |  lr | epochs | memory(MB) | RTX3090 hours |
> :-:|:-:|:-:|:-:|:-:|:-:
> FR|256|0.0001|300|3504|0.21
> Inter_RAT|256|0.001|20|3660|2.34
> MCD|128|0.0001|150|2630|0.28
>
> Since the memory usage is affected by the batchsize, we further assign our MCD with a batchsize of 256 (the same as FR and Inter_RAT), and the memory usage then becomes 3806 MB.
> Since our approach is straightforward and does not introduce additional regularizers, we have not identified specific modules that require dedicated ablation. However, Table 4 is a synthetic experiment crafted to demonstrate MCD's efficacy against degeneration, it can be considered somewhat of an ablation study (where feature correlations are ablated). In the future, we will consider integrating our MCD criterion with other sophisticated methods and conducting appropriate ablations.
>
> **A8 (why bert doesn't work well):** This could be a combination of several complex reasons. Here are some possible factors: First, fine-tuning large models is challenging in itself, especially when there is not much data available. Second, most of previous methods are  very sensitive to hyperparameter tuning (implied in the A2R paper), which could lead to training instability and result in a higher likelihood of getting stuck in local optima. Third, the more powerful the explainer is, the more easier degeneration will be (implied in a recent paper [A]).
>
> We agree that it would be valuable to explore specifically what happens with BERT It's somewhat beyond the scope of this paper, and we leave it as future work.
>
> **A9 (Limitations):** We appreciate your suggestions, and we will certainly expand on these points and discuss these limitations in more detail in the next version.
>
> [A] Unsupervised Selective Rationalization with Noise Injection. ACL 2023.

---

> > ### Comment · Reviewer_Rbzy · 2023-08-18
> >
> > Thank you. These are reasonable responses, and I still am comfortable with my rating.

---

> > > ### Author Response · Authors · 2023-08-21
> > >
> > > Thank you very much for taking the time to review our paper! With best wishes to you and yours!

---

### Official Review · Reviewer_xX9m · 2023-07-11

**Soundness:** 3 good
**Presentation:** 3 good
**Contribution:** 3 good
**Rating:** 7
**Confidence:** 3

**Summary:**

This work focuses on the problem of rationalization that aims to extract explanatory sentences that serve as rationales along with training a predictor for the downstream task. Prior work on rationalization has typically employed the maximum mutual information (MMI) criterion. However, this criterion does not uncover causal relationships and latches on to feature correlations and degenerations. The key insight in this work is that non-causal features are independent of the target labels given the causal features, based on which the authors propose a minimum conditional dependence criterion to discover causal rationales.

**Strengths:**

- Paper is written well.
- Offer a unified perspective of feature correlations and degenerations (described in prior work on rationalization).
- Propose a new MCD criterion to identify causal rationales.
- Outperform existing MMI-based rationalization techniques on two multi-aspect sentiment classification tasks.

**Weaknesses:**

One of the main contributions of this work is that the proposed MCD criterion leads to the discovery of causal rationales. It would be useful for the reader to see some examples of generated rationales via MCD, and also highlight failure cases when MCD fails to retrieve the causal rationales.

**Questions:**

- Why is Equation 16 a good approximation? Define \Omega(M) in Equation 16.
- In Table 5, why are the F1 scores of FR-ELECTRA so poor on the Appearance and Palate aspects?
- Some anecdotal examples of causal rationales that were identified via MCD as opposed to MMI would be useful for the reader.

**Limitations:**

Some limitations have been listed.

---

> ### Author Rebuttal · Authors · 2023-08-09
>
> Thank you deeply for taking the time to thoroughly review our paper. We are truly grateful for the insights and recommendations you've provided.
>
> **Q1:** It would be useful for the reader to see some examples of generated rationales via MCD, and also highlight failure cases when MCD fails to retrieve the causal rationales.
>
> **A1:** We appreciate your suggestion a lot. We have now added some examples and failure cases in Figure 1 and 2 of the rebuttal.pdf.
>
> In the example in Figure 1, the causal rationale is the text describing the good aroma of the beer. We see that RNP selects only the text describing the Palate aspect. Inter_RAT selects part of Aroma, but also Palate (e.g., "felt very smooth"). FR also selects both Aroma and Palate. Our MCD selects only Aroma.
>
> In the failure case in Figure 2, we observe that MCD effectively identifies simpler sentiment-oriented descriptions related to aroma but struggles to recognize higher-level logical reasoning. For instance, expressions like "the first one that has been" often carry evident emotional inclinations, but such phrases are not selected by MCD due to their reliance on common sense and logical reasoning.
>
> **Q2:** Why is Equation 16 a good approximation? Define \Omega(M) in Equation 16.
>
> **A2:** As you know, the cross-entropy can be expanded as follows:
> $H_c(Y,\hat{Y}|X)=H(Y|X)+D_{KL}(P(Y|X)||P(\hat{Y}|X))$.
> When we get the minimum cross-entropy $H_c(Y,\hat{Y}|X)$, we have $D_{KL}(P(Y|X)||P(\hat{Y}|X))=0$, which means $P(Y|X)=P(\hat{Y}|X)$. In this case, $P(\hat{Y}|X)$ is a good approximation for $P(Y|X)$. And it's the same for how $P(Y|X_Z)$ is approximated by $P(\hat{Y}|X_Z)$.
> As for $\Omega(M)$, it's defined in Equation 4 (Line 154). We appreciate your suggestion and agree that it will be good for the readers if we imply it again in Equation 16, and we will add it in the next version.
>
> **Q3:** In Table 5, why are the F1 scores of FR-ELECTRA so poor on the Appearance and Palate aspects?
>
> **A3:** Thank you for your question. In fact, we think we should approach this phenomenon from the opposite perspective, that is, why FR performs well on the Aroma aspect. As shown in Table 6, we see that most previous methods perform very poorly when conducted with over-parameterized BERT. On the Appearance aspect, our reimplementation of FR is better than the one reported by FR itself (the results in Table 6 are reported by FR itself).
> So why can FR perform well on the Aroma aspect? We are not sure, but we note that a recent paper called CR [A], publised on ICML 2023, shows similar results:
> |      F1     | Appearance | Aroma | Palate |
> |:-----------:|:----------:|:-----:|:------:|
> | CR-BERT     | 28.0       | 39.0  | 26.5   |
> | FR-ELECTRA  |    18.0    |  56.7 |  11.3  |
>
> We see that CR also performs well on Aroma, but poorly on Appearance and Palate. The reason may be that Aroma is a relatively easy aspect and the other two aspects are relatively harder. However, a more precise understanding of the underlying mechanism may be highly complex, which to some extent goes beyond the scope of this paper.
>
> [A] Towards Trustworthy Explanation: On Causal Rationalization. ICML 2023.

---

> > ### Comment · Reviewer_xX9m · 2023-08-19
> > **Response to rebuttal**
> >
> > Thanks to the authors for their detailed response (including new results) and clarifications to my questions. I'm raising my score to 7.

---

> > > ### Author Response · Authors · 2023-08-21
> > >
> > > Thank you very much for taking the time to review our paper! With best wishes to you and yours!

---

### Author Rebuttal · Authors · 2023-08-09

We are deeply grateful to every reviewer for their in-depth analysis and constructive feedback on our manuscript.

Here are the figures and tables of some of the new experimental results, attached to rebuttal.pdf.

---

### Decision · Program_Chairs · 2023-09-21

**Decision:**

Accept (poster)

**Comment:**

The method proposed to discover causal rationales is interesting as acknowledge by two reviewers. However, one reviewer with a high confidence remained unconvinced on couple of points. Considering the paper's merits, I am in favor of accepting the paper.